# Decoder-Hybrid-Decoder Architecture for Efficient Reasoning with Long Generation

**Liliang Ren**[1]    **Congcong Chen**[1]    **Haoran Xu**[1]    **Young Jin Kim**[1]
**Adam Atkinson**[1]    **Zheng Zhan**[1]    **Jiankai Sun**[2]    **Baolin Peng**[1]
**Liyuan Liu**[1]    **Shuohang Wang**[1]    **Hao Cheng**[1]
**Jianfeng Gao**[1]    **Weizhu Chen**[1]    **Yelong Shen**[1]

[1]Microsoft    [2]Stanford University

{liliangren,yeshe}@microsoft.com

## Abstract

Recent advances in language modeling have demonstrated the effectiveness of State Space Models (SSMs) for efficient sequence modeling. While hybrid architectures such as Samba and the decoder-decoder architecture, YOCO, have shown promising performance gains over Transformers, prior works have not investigated the efficiency potential of representation sharing between SSM layers. In this paper, we introduce the Gated Memory Unit (GMU), a simple yet effective mechanism for efficient memory sharing across layers. We apply it to create SambaY, a decoder-hybrid-decoder architecture that incorporates GMUs in the cross-decoder to share memory readout states from a Samba-based self-decoder. SambaY significantly enhances decoding efficiency, preserves linear pre-filling time complexity, and boosts long-context performance, all while eliminating the need for explicit positional encoding. Through extensive scaling experiments, we demonstrate that our model exhibits a significantly lower irreducible loss compared to a strong YOCO baseline, indicating superior performance scalability under large-scale compute regimes. Our largest model enhanced with Differential Attention, Phi4-mini-Flash-Reasoning, achieves significantly better performance than Phi4-mini-Reasoning on reasoning tasks such as Math500, AIME24/25, and GPQA Diamond without any reinforcement learning, while delivering up to 10× higher decoding throughput on 2K-length prompts with 32K generation length under the vLLM inference framework. We release our training codebase on open-source data at https://github.com/microsoft/ArchScale.

## 1 Introduction

State Space Models (SSMs) [GGR21, GGGR22, GD23, DG24], including linear attention [HDLL22, SDH$^+$23, QHS$^+$22, YWS$^+$23, YWZ$^+$24, YKH25] and modern Recurrent Neural Networks (RNNs) [BPS$^+$24, BPL$^+$25, PAA$^+$23, GOA$^+$24], have recently shown promising results for more efficient sequence modeling over Transformers [VSP$^+$17]. While pure SSMs/RNNs offer computational advantages with their linear complexity, they exhibit a theoretical expressiveness gap on the in-context retrieval capability relative to Transformers [WDL24]. This gap can be bridged by hybrid architectures [LLB$^+$24, DSF$^+$24, RLL$^+$25, WBR$^+$24, DFD$^+$25, Min25], even with the inclusion of as few as a single full attention layer [WDL24]. Recently, the decoder-decoder architecture, YOCO [SDZ$^+$24], achieves linear complexity for long context processing through storing the Key-Value (KV) pairs from a single self-attention layer and re-using them across all subsequent layers through cross-attentions. In practice, YOCO has delivered substantial efficiency gains when processing user prompts with long sequences, but challenges remain; it does not mitigate the attention memory

39th Conference on Neural Information Processing Systems (NeurIPS 2025).

I/O cost for its cross-attentions during the generation stage of the model responses. This limitation becomes particularly pronounced for modern large language models (LLMs) [Ope24, DA25] that generate extensively long Chains-of-Thought (CoTs) [WWS+22] for hard reasoning tasks.

In this paper, we investigate the potential of representation sharing between SSM layers to enhance decoding efficiency. We introduce the Gated Memory Unit (GMU), a versatile, simple yet effective mechanism for efficient memory sharing across layers. Applying GMUs to the cross-decoder of YOCO, we create a novel model with our decoder-hybrid-decoder architecture named SambaY that uses Samba [RLL+25] for the self-decoder and replaces half of the cross-attention layers with GMUs to share the inner representations of the final SSM layer in the self-decoder. Since around 50% of expensive cross-attention layers are replaced with cheap element-wise gating, SambaY significantly improves decoding efficiency and maintains a linear pre-filling time complexity, all while removing the need for explicit positional encoding such as RoPE [SLP+21].

To enable a robust comparison of the scaling capabilities across different architectures, we first design a principled $\mu$P++ hyperparameter transfer scheme that accounts for both depth and width scaling, as well as the application of weight decay to vector-like parameters. We then conduct extensive experiments up to 3.4B parameters/600B tokens to verify the scaling behaviors of both our $\mu$P++ scaling laws and the SambaY architecture. Comparing to Samba+YOCO, an architecture that naively combines Samba with YOCO, we show that SambaY has significantly lower irreducible loss [HNA+17] on the validation set when scaling with the training FLOPs, indicating a better scaling potential with large-scale computes. We also conduct extensive experiments to verify the long-context retrieval capabilities of our architecture. Our results reveal that SambaY achieves superior performance on challenging long-context tasks like Phonebook and RULER [HSK+24] benchmark, even with a modest Sliding Window Attention (SWA) size of 256. To further explore the capabilities of hybrid models with a single set of full attention memory, we augment SambaY with Differential Attention [YDX+24], resulting in the Phi4-mini-Flash architecture. We pre-train our 3.8B-parameter model Phi4-mini-Flash with 5T tokens from the same Phi4-mini data corpus and further follow Phi4-mini-Reasoning [XPA+25] to conduct the multi-stage distillation with Supervised Fine-Tuning (SFT) and Direct Preference Optimization (DPO) to produce our reasoning model, Phi4-mini-Flash-Reasoning. Our model achieves significantly better performance than the strong Phi4-mini-Reasoning baseline on challenging reasoning benchmarks such as Math500, AIME24/25, and GPQA Diamond, while excluding any stage of Reinforcement Learning (RL) that is used by Phi4-mini-Reasoning. Critically, our Phi4-mini-Flash-Reasoning delivers up to $10\times$ higher decoding throughput on 2K-length prompts with 32K generation length under the vLLM [KLZ+23] inference framework, showcasing its substantial and practical efficiency gains for the LLM reasoning paradigm of generating long Chain-of-Thoughts.

## 2 Decoder-Hybrid-Decoder Architecture

Inspired by the gating mechanism that broadly exists in Gated Linear Units [Sha20], Gated Attention Units [HDLL22] and SSMs [GD23, YWS+23, YKH25], we introduce our Gated Memory Unit (GMU) together with its application on YOCO, which produces our final decoder-hybrid-decoder architecture. We include a dedicated related works section in Appendix I, the limitation section in Appendix J and provide the background introduction of YOCO in Appendix A.

**Token mixing as a matrix operator.** Both state-space models (SSMs) and self-attention layers perform token mixing through a linear operator that can be written as a matrix $A \in \mathbb{R}^{n \times n}$, where $n$ is the sequence length. In SSMs, $A$ is a highly structured matrix that captures the parallel form of an underlying recurrent update, whereas in self-attention $A$ is the row-aggregating attention matrix whose entries are the query-key softmax probabilities. For a given head at layer $l'$, the mixed representation is

$$M^{(l')} = A^{(l')} V^{(l')},$$

where $V^{(l')} \in \mathbb{R}^{n \times d_h}$ denotes either the SSM state inputs or the attention value vectors.

**Gated Memory Unit (GMU).** From an inter-layer perspective, we define "memory" as hidden representations passed from preceding layers. Specifically, at a given layer $l$, GMU operates on two inputs: the current layer's input hidden state, $X_l \in \mathbb{R}^{n \times d_m}$, and the mixed representation $M^{(l')} \in \mathbb{R}^{n \times d_h}$ of a previous layer $l'$ (where $l' < l$). The GMU then produces an output $Y_l \in \mathbb{R}^{n \times d_m}$

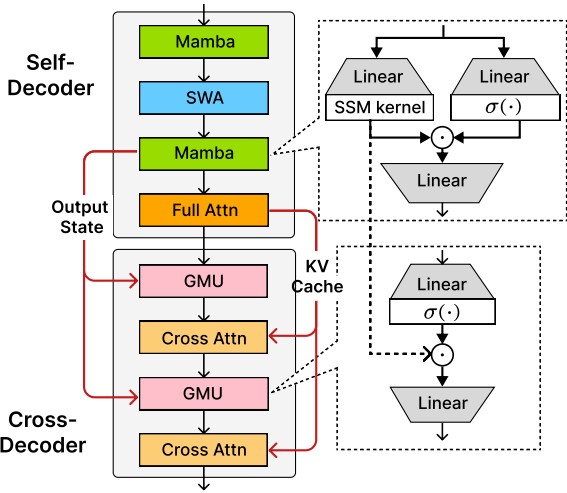

Figure 1: Our decoder-hybrid-decoder architecture taking Samba [RLL+25] as the self-decoder. Gated Memory Units (GMUs) are interleaved with the cross-attention layers in the cross-decoder to reduce the decoding complexity. As in YOCO [SDZ+24], the full attention layer only need to compute the KV cache during prefilling with the self-decoder, leading to linear computation complexity for the prefill stage.

through a gating mechanism modulated by learnable projections. Formally, the GMU can be expressed as:

$$Y_l = (M^{(l')} \odot \sigma(X_l W_1^T))W_2$$

where $\sigma(\cdot)$ is the SiLU [EUD17] activation function, $\odot$ denotes element-wise multiplication, and $W_1, W_2 \in \mathbb{R}^{d_h \times d_m}$ are learnable weight matrices. One can also apply an RMSNorm [ZS19] layer after the element-wise multiplication, yielding the normalized GMU (nGMU). We show that this normalization is crucial when the memory originates from linear attention mechanisms [DG24, YKH25], as detailed in Appendix B and Appendix H. Intuitively, the GMU enables a dynamical and fine-grained recalibration of the token mixing performed in a previous layer, conditioned on the current layer's input across each of the memory channels. Specifically, for each element $H_{ik}$ in the gated output $H = M^{(l')} \odot G^{(l)} \in \mathbb{R}^{n \times d_h}, G^{(l)} = \sigma(X_l W_1^T)$ , we have

$$H_{ik} = G_{ik}^{(l)} \sum_j A_{ij}^{(l')} V_{jk}^{(l')} = \sum_j G_{ik}^{(l)} A_{ij}^{(l')} V_{jk}^{(l')} = \sum_j \underbrace{A_{ij}^{(l')} G_{ik}^{(l)}}_{\tilde{A}_{ijk}} V_{jk}^{(l')},$$

which shows that the gate $G^{(l)}$ effectively lifts the previous token mixing matrix $A^{(l')}$ into a third-order tensor with elements $\tilde{A}_{ijk} = G_{ik}^{(l)} A_{ij}^{(l')}$, yielding a learned, channel-specific reweighting of the original token-mixing operator while maintaining linearity on the original value matrix $V^{(l')}$ at a later layer $l$. The concept of the GMU generalizes beyond token-mixed memories. For instance, it can gate intermediate outputs from previous MLP layers, enabling retrieval from a projected representation of an earlier layer's input. In all cases, GMUs reduce both parameter count and computational cost compared to standard SSM, attention, or MLP layers.

**Model architecture.** In Figure 1, we illustrate the SambaY architecture, using our decoder-hybrid-decoder architecture with Samba [RLL+25] as the self-decoder. We apply GMUs to the cross-decoder of YOCO to replace half of its cross-attention layers. The GMUs share the representation from the last SSMs layers in the self-decoder so that the pre-filling time complexity is still linear. Compared to YOCO, our approach only requires caching an additional SSM kernel output state $\mathbf{m} \in \mathbb{R}^{d_h}, d_h = 2d_m$ from the final Mamba layer, an overhead that is negligible in size, alongside the KV cache from the last full-attention layer during pre-filling. During decoding, we reduce the memory I/O complexity for half of the cross-attention layers from a linear cost of $O(d_{kv}N)$ to a constant $O(d_h)$, where $N$ is the sequence length and $d_{kv}$ is the dimension of key/value vectors. This leads to significant efficiency gains when $N \gg d_h/d_{kv}$, a condition that is easily met in practice since the ratio $d_h/d_{kv}$ typically does not exceed 128.

# 3 Experiments and Results

Motivated by the theoretical efficiency of our SambaY architecture, we aim to address the following research questions: Does the architecture scale effectively? Does it compromise long-context performance? Can it support reasoning over long generations? Given that a neural architecture's performance is tightly coupled with its optimization and initialization settings, we begin by establishing a generic scaling setup to encourage a fair comparison of scaling behavior across different architectures.

**Baseline Architecture.** Apart from our proposed SambaY architecture, we consider the following baselines in this section: Transformer++ [HDLL22] (which uses SwiGLU [Sha20] for MLP and RoPE [SLP$^+$21] with the base frequency of 10,000), Samba+YOCO (which uses Samba as self-decoder for the original YOCO architecture), SWA+YOCO (the original YOCO with SWA layers as self-decoder), TransformerLS (interleaving SWA with full attention using a layer ratio of 3:1), and SambaY+DA (which uses Differential Attention (DA) [YDX$^+$24] for all attention layers in SambaY). More architecture details are included in Appendix D. We standardize the sliding window size to 128 for all architectures with SWA while conducting extensive ablations on window size effects in Section 3.2. Following the studies in recent hybrid models [LLB$^+$24, RLL$^+$25], we omit explicit positional encodings (NoPE) for all hybrid SSMs architectures, and demonstrate that NoPE enables zero-shot $2\times$ retrievable context extrapolation in Appendix F.

## 3.1 Scaling Experiments on Open-Source Data

**Architecture scaling setup.** We use a simple linear rule from the previous works on Transformer models [KMH$^+$20, TJY$^+$24] for scaling the architectural shape of our Transformer++ baseline, including model width $w$, model depth $d$, number of attention query heads $h_q$ and the MLP inner dimension $w_{mlp}$, *i.e.*,

$$w = \alpha d, \quad \alpha = \alpha_0 = 128, \quad h_q = d, \quad h_{kv} = d/4, \quad w_{mlp} = 4w,$$

where the Transformer-specific aspect ratio $\alpha_0$ and the number of key-value heads $h_{kv}$ are computed based on the Llama-3-8B [Met24] architecture. The total number of non-embedding parameters $N(d)$ for the Transformer++ architecture can then be calculated as,

$$N_{\text{attn}}(d) = 2.5dw^2, N_{\text{mlp}}(d) = 12dw^2,$$

$$N(d) = N_{\text{attn}}(d) + N_{\text{mlp}}(d) = 14.5dw^2 = 237568d^3,$$

where $N_{\text{attn}}, N_{\text{mlp}}$ means the total number of parameters for attention/MLP layers respectively.

**Scaling transfer for hybrid architectures.** Since different token mixers have their own inner dimension expansion ratio, it is hard to balance the number of parameters between hybrid models and Transformers to make fair comparisons. Previous works [DA24a, RLL$^+$25, YKH25] often adjust the model depth to tie the total number of parameters, but this could change the memory cache size significantly (e.g. adding two attention layers in a 12-layer Transformer resulting in a 16.7% increase of KV cache size), making unfair comparisons on the inference time cost. We propose a simple solution that (1) builds an iso-parametric equation with respect to the aspect ratio via aligning the total number of non-embedding parameters to the Transformer baseline, (2) solves the equation to obtain the specific aspect ratio for the hybrid architectures. We maintain consistent hyperparameter settings with a $4\times$ MLP inner dimension expansion ratio and GQA [ALTdJ$^+$23] group size of 4 for self-attention layers, matching our Transformer++ baseline. We also fix the head dimension to be $\alpha_0 = 128$, and set the inner dimension of the attention layers to be $w_{\text{attn}} = \alpha_0 d$ so that the number of key-value heads $h_{kv}$ is a valid integer. Specifically, for SambaY, we have

$$N_{\text{attn}}(d) = 2.5dw \cdot w_{\text{attn}}/4 + 2dw \cdot w_{\text{attn}}/4, \quad N_{\text{mamba}}(d) = 6dw^2/4, \quad N_{\text{gmu}}(d) = 4dw^2/4,$$

$$N(d) = N_{\text{attn}}(d) + N_{\text{mamba}}(d) + N_{\text{mlp}}(d) + N_{\text{gmu}}(d) = 144\alpha d^3 + 14.5\alpha^2 d^3 = 237568d^3,$$

where $N_{\text{attn}}, N_{\text{mamba}}, N_{\text{mlp}}, N_{\text{gmu}}$ means the total number of parameters for attention/Mamba/MLP/GMU layers. Solving for $\alpha$, we get $\alpha_1 \approx 124$. For Samba+YOCO, we can similarly solve an iso-parametric equation to obtain $\alpha_2 \approx 126$, with more details in Appendix C.

**Hyperparameter scaling with $\mu$P++.**  To account for both width and depth scaling of model architectures, we propose $\mu$P++ hyperparameter scaling laws that integrate $\mu$P [YHB$^+$22], Depth-$\mu$P [YYZH23], and apply zero weight decay to vector-like or scalar-like parameters[1] for training stability. Since we use the AdamW optimizer [LH18], we apply batch-size-based scaling with $\eta \propto \sqrt{B}$. The learning rate is further scaled as $\eta \propto 1/\sqrt{d}$ following Depth-$\mu$P. For studying the FLOPs scaling behavior across model architectures, we adopt the Chinchilla scaling law [HBM$^+$22] to scale the number of training tokens $T$ linearly with the number of model parameters. Formally, we have

$$\eta = \eta_0 \sqrt{\frac{Bd_0}{B_0 d}}, \quad B = B_0, \quad T = T_0 \frac{N(d)}{N(d_0)},$$

where the base learning rate is set as $\eta_0 = 4 \times 10^{-4}$ and the base batch size $B_0 = 2^{21} = 2\text{M}$ number of tokens. We also explore scaling the batch size sub-linearly with respect to the training tokens [MKAT18, SWW$^+$24, LZH$^+$25], but find that it harms the data scaling behavior of the models, so we keep the batch size as a constant across scales. The base model depth is set as $d_0 = 16$ so that $N(d_0) \approx 10^9$ number of parameters. The base training token count $T_0$ is set to 100B, corresponding to $5\times$ the Chinchilla-optimal ratio of tokens per parameter (approximately 20 based on [HBM$^+$22]), in order to study scaling behaviors in a typical over-training regime. We summarize the differences between Standard Parametrization, $\mu$P and $\mu$P++ in Appendix D, while providing large scale ablation studies in Appendix E.

**Scaling experiment setups.**  A common concern with SSMs is that they are not theoretically more expressive than self-attention for in-context retrieval [WDL24]. This raises the question of whether the better performance of hybrid SSM models is owing to their fast convergence from the recency bias, while Transformers could eventually match their performance given more training tokens. With the scaling laws we established in the previous paragraphs, we can now examine this hypothesis systematically. We first study the data scaling behavior across architectures through fixing the model size at 1B parameters with $d = 16$ and scaling the number of training tokens $T$ from 100B to 600B. We also study the FLOPs scaling behaviors of the model architectures with up to 3.4B parameters and 342B tokens through varying the model depth $d = \{8, 12, 16, 20, 24\}$. We use a 4K training sequence length and the SlimPajama [SAKM$^+$23] dataset for all our scaling experiments. More experimental details are included in Appendix D.

**Comparison of scaling behaviors.**  To quantitatively compare the scaling trajectories, we fit the validation loss $L$ as a function of compute (FLOPs), denoted as $D_{\text{FLOPs}}$, to a power law [HNA$^+$17, HBM$^+$22] of the form:

$$L(D_{\text{FLOPs}}) = A \cdot D_{\text{FLOPs}}^{-b} + C$$

This model enables us to estimate the irreducible loss $C$ which represents the lower bound of achievable loss for a given architecture or parameterization under infinite compute, and the scaling exponent $b$ that reflects the learning efficiency with respect to compute. We fit the curves with least squares and the LMA algorithm [LEV44, Mar63]. A similar power law model is employed for data scaling experiments, where loss is modeled as a function of the number of training tokens $D_{\text{tokens}}$.

In Figure 2, we present the results of both FLOPs scaling and data scaling experiments, showing validation loss on the SlimPajama dataset as a function of total training FLOPs or number of training tokens. We show both the original data points and the fitted power-law curves. The goodness of fit for each curve is assessed using the $R^2$ statistic and all plots have a fitness score $R^2 \geq 0.999$, indicating near-perfect fits. While larger values of the scaling exponent $b$ or the coefficient $A$ indicate that a model may converge more rapidly given a small-scale compute or data budget, these parameters alone do not necessarily predict superior performance at larger scales. Therefore, we emphasize the irreducible loss $C$ obtained from scaling law fitting as the primary metric for assessing an architecture's long-term scaling potential. As illustrated in Figure 2a, the SambaY architecture exhibits the lowest irreducible loss ($C = 0.58$) for FLOPs scaling, suggesting that it can attain a superior validation loss when scaled further with substantially increased computational resources. However, under $\mu$P++, all architectures share the same compute efficiency exponent ($b = 0.07$), indicating that the hybrid architectures explored did not yield improvements in models' learning

---

[1]Following the definition in $\mu$P, parameters are vector-like when exactly one dimension scales with model width (e.g., embedding and unembedding layers), and scalar-like when no dimension scales with width.

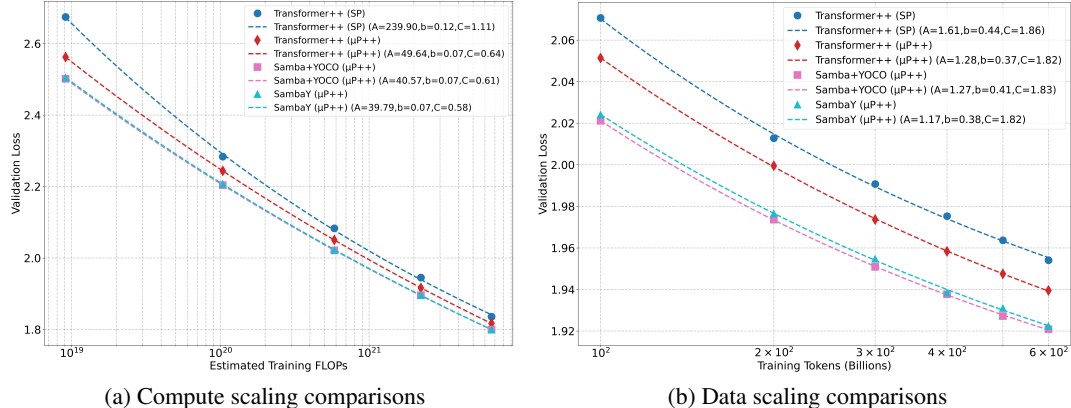

(a) Compute scaling comparisons      (b) Data scaling comparisons

Figure 2: Validation Loss v.s. FLOPs (left) or Training Tokens (right) on the SlimPajama dataset. Besides the architecture comparisons, we also compare our $\mu$P++ based scaling with the Standard Parametrization (SP).

efficiency with respect to compute. We also observe that $\mu$P++ yields a lower irreducible loss than SP under both data and compute scaling, indicating more favorable scaling potentials.

Notably in Figure 2b, the Transformer++ model trained with $\mu$P++ exhibits a large validation loss gap compared to SambaY and Samba+YOCO within the measured range of training tokens. However, its fitted irreducible loss ($C = 1.82$) is nearly identical to those of the hybrid models, suggesting that with an infinite amount of data, Transformer++ can eventually catch up—albeit with slower convergence. This aligns with our expectation, as there is no theoretical expressiveness gap between Transformers and our hybrid models when the number of parameters is held constant. This can be because we use Mamba-1 as our SSM which falls into the same complexity class of $TC^0$ as Transformers [MPS24]. Interestingly, this convergence no longer holds when both model size and data scale proportionally. As illustrated in Figure 2a, under the $\mu$P++ setting, Transformer++ exhibits the highest irreducible loss $C = 0.64$, indicating that hybrid architectures could offer superior scalability under limited data regimes.

## 3.2 Efficient Long Context Retrieval

Given the presence of full-attention layers, we aim to determine the minimal sliding window size required for our hybrid models to retain effective long-context retrieval capabilities—an essential property for supporting advanced reasoning behaviors that involve long generation with backtracking. Specifically, we pre-train 1.0B parameter models with $\mu$P++ and $d = 16$ using TransformerLS, SambaY, SambaY+DA and Samba+YOCO architectures respectively on the ProLong-64k [GWYC24] dataset with 32K sequence length and 40B tokens, varying the window size of their Sliding Window Attention (SWA) in the range $\{64, 128, \ldots, 2048\}$. We align the number of parameters between different architectures through building the iso-parametric equations as in Section 3.1. We adopt variable-length training, where short documents are packed together and self-attended within the same seg-

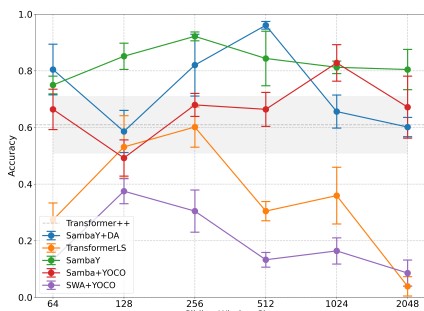

Figure 3: Accuracy (with error bars) v.s. Sliding Window Size on Phonebook with 32K evaluation length.

ment. We evaluate the long-context retrieval capabilities of the models using a difficult Phonebook benchmark [JBKM24] with a 32K context length (containing 1,850 name-number pairs). We choose this benchmark because it is a realistic multi-key-value retrieval task with minimal instructions, which minimizes the confounding influence of instruction-following ability when evaluating long-context retrieval performance. We use a RoPE base of 640K for TransformerLS, Transformer++ and SWA+YOCO, following the lower bounds proposed in [XMW+24]. We also examine how the training corpus and methods affect the long context performance, with more details in Appendix F.

Table 1: Retrieval accuracy on Needle-In-A-Haystack (NIAH) tasks with 32K context from the RULER [HSK+24] long context benchmark. MK: Multi-Key, MQ: Multi-Query, MV: Mutli-Value, S: Single-needle. We use the best Sliding Window Attention (SWA) size found on the Phonebook benchmark for each hybrid architecture. Best results are in bold, second best underlined.

| Model | SWA | MK-1 | MK-2 | MK-3 | MQ | MV | S-1 | S-2 | S-3 | Avg. |
|---|---|---|---|---|---|---|---|---|---|---|
| Transformer++ | - | 36.4 | 3.8 | 0.0 | 27.9 | 24.1 | 94.8 | 66.0 | 31.0 | 35.5 |
| TransformerLS | 256 | 42.8 | 6.0 | 0.0 | **29.8** | **27.5** | 91.8 | 49.6 | 23.4 | 33.9 |
| SWA+YOCO | 128 | 24.2 | 6.8 | 0.2 | 10.2 | 14.7 | 81.2 | 32.6 | 48.4 | 27.3 |
| Samba+YOCO | 1024 | 49.0 | **28.0** | **2.6** | 12.8 | 18.3 | **100.0** | 63.2 | 23.6 | 37.2 |
| SambaY | 256 | 54.6 | 27.8 | 0.4 | 12.7 | 19.4 | 83.2 | 81.2 | 63.8 | 42.9 |
| SambaY+DA | 512 | **64.6** | 27.6 | 0.2 | 12.8 | 19.9 | 99.8 | **86.4** | **69.6** | **47.6** |

As shown in Figure 3, which plots accuracy against SWA size on the Phonebook dataset (32K evaluation length), surprisingly, larger SWA sizes do not consistently provide better results. Since learned full attention involves both sliding window (local) patterns and non-local patterns like global retrieval or attention sinks, using small sliding window sizes, where models like SambaY and SambaY+DA show strong performance, could enable the model to focus on local patterns more easily and mitigate issues like attention sinks [GPD+25]. Moreover, shorter sliding windows can facilitate faster convergence, a crucial factor in long context training scenarios that are often characterized by limited high-quality data. The lower scores of SWA+YOCO, which consistently underperform the SambaY variants, could indicate that pure attention-based models require more substantial data for long-context training.

Table 2: Downstream short-context evaluation on language modeling and common-sense reasoning tasks in zero-shot for 1B-parameter models with the tuned sliding window size. The training speed is measured in MTPS (Million Tokens Per Second) with 64 A100-80GB GPUs. Best results are in bold, second best underlined.

| Model | SWA | Speed mtps ↑ | Wiki. ppl ↓ | LMB. acc ↑ | ARC-c acc_n ↑ | ARC-e acc ↑ | Hella. acc_n ↑ | PIQA acc ↑ | Wino. acc ↑ | Avg. acc ↑ |
|---|---|---|---|---|---|---|---|---|---|---|
| Transformer++ | - | 0.89 | 19.75 | 45.45 | 27.56 | 54.17 | 43.86 | 68.77 | 50.28 | 48.35 |
| TransformerLS | 256 | **1.46** | 18.49 | 48.77 | 28.84 | 57.11 | 45.85 | 69.21 | 53.67 | 50.57 |
| SWA+YOCO | 128 | 1.24 | 18.01 | 49.80 | 28.58 | 57.79 | 46.48 | 70.46 | 51.85 | 50.69 |
| Samba+YOCO | 1024 | 0.99 | 16.73 | **50.53** | 28.50 | 60.02 | 48.85 | **71.55** | 52.57 | 52.00 |
| SambaY | 256 | 1.11 | 17.83 | 50.40 | **29.44** | 57.87 | 49.08 | 71.00 | **55.25** | **52.17** |
| SambaY+DA | 512 | 0.91 | **16.59** | 49.68 | 28.33 | **60.65** | 49.53 | 71.38 | 53.43 | **52.17** |

Using the optimal sliding window size from the Phonebook benchmark, we evaluate our architectures on both long-context retrieval tasks (Table 1) and traditional downstream benchmarks (Table 2). Across both contexts, hybrid models with SSMs consistently outperform pure Transformer architectures. SambaY variants demonstrate notable advantages in long-context retrieval while maintaining strong performance on short-context tasks, despite using much smaller sliding window sizes than Samba+YOCO. The addition of DA further enhances multi-key and single-needle retrieval capabilities, while Transformer-based models show specific strengths in multi-query and multi-value scenarios. TransformerLS/SWA+YOCO outperforms Transformer++ on short-context tasks but falls behind on the RULER benchmark, highlighting the trade-off on long-context performance caused by introducing SWA to full attention models. Overall, our results suggest that GMUs facilitate efficient representation sharing across layers and enable strong performance with smaller SWA sizes.

## 3.3 Large-Scale Pre-training on High-quality Data

We apply our SambaY+DA architecture to pre-train a larger-scale prototype model named Phi4-mini-Flash with 3.8B parameters. It uses an SWA size of 512 and GQA of group size 2. Compared to the configuration described in Section 3.1, this model uses a different aspect ratio $\alpha = 80$ and an attention head dimension of 64. It is trained with standard parameterization rather than $\mu$P++ due to resource constraints at the time of scaling study. We pre-train our model on 5T tokens from the data corpus used by Phi4-mini [MAA+25] on 1K A100-80GB GPUs for 14 days. During training, we encounter severe loss divergence, which we mitigate by introducing label smoothing of 0.1 and

Table 3: Downstream evaluation performance of post-trained models. We use the completion split for BigCodeBench evaluation. Bold indicates the best result per row.

| Benchmark | Metric | Phi4-mini | Phi4-mini-Flash |
|---|---|---|---|
| MMLU [HBB+21] | 5-shot | 67.3 | **71.9** |
| MMLU-Pro [WMZ+24] | 0-shot, CoT | 52.8 | **54.7** |
| Arena Hard [LCF+24] | Win Rate | 32.8 | **34.9** |
| GSM8K [CKB+21] | 0-shot, CoT | 88.6 | **89.5** |
| Qasper [DLB+21] | F1 | **40.4** | 40.2 |
| SummScreenFD [CCWG22] | ROUGE-L | 16.0 | **17.0** |
| BigCodeBench [ZVC+25] | pass@1 | 43.0 | **44.5** |
| MBPP [AON+21] | pass@1 | 65.3 | **69.8** |

attention dropout of 0.05. The optimization setup here is by no means optimal, as the primary goal of this experiment is to evaluate the viability of our architecture at larger scales. Phi4-mini-Flash uses a 200K token vocabulary size consistent with Phi4-mini. Additional training and architectural details, including the mitigation of stability issues, are provided in Appendix D. Table 3 demonstrates that Phi4-mini-Flash outperforms the Phi4-mini baseline across a diverse range of tasks, with notable improvements on knowledge-intensive benchmarks like MMLU and coding tasks such as MBPP. The consistent performance advantage, winning on 7 out of 8 benchmarks, is particularly significant given that Phi4-mini-Flash achieves these gains while maintaining substantially higher computational efficiency during inference.

## 3.4 Efficient Reasoning with Long Generation

Table 4: Pass@1 performance of models on reasoning benchmarks measured with a maximum generation length of 32K. We report Pass@1 accuracy averaged over 64 samples for AIME24/25 and 8 samples for Math500 and GPQA Diamond to ensure evaluation robustness. We also evaluate popular open-source distilled reasoning models [DA25, Bes, Ope] as reference baselines.

| Model | AIME24 | AIME25 | Math500 | GPQA Diamond |
|---|---|---|---|---|
| DeepSeek-R1-Distill-Qwen-1.5B | 29.58 | 20.78 | 84.50 | 37.69 |
| DeepSeek-R1-Distill-Qwen-7B | 53.70 | 35.94 | 93.03 | 47.85 |
| DeepSeek-R1-Distill-Llama-8B | 43.96 | 27.34 | 87.48 | 45.83 |
| Bespoke-Stratos-7B | 21.51 | 18.28 | 80.73 | 38.51 |
| OpenThinker-7B | 29.69 | 24.32 | 87.25 | 41.60 |
| Phi4-mini-Reasoning (3.8B) | 48.13 | 31.77 | 91.20 | 44.51 |
| Phi4-mini-Flash-Reasoning (3.8B) | **52.29** | **33.59** | **92.45** | **45.08** |

Our Phi4-mini-Flash-Reasoning model is continually trained from the Phi4-mini-Flash model with the same multi-stage distillation data following Phi4-mini-Reasoning [XPA+25]. Due to the limited resources, we only conduct the distillation with SFT and DPO stages and leave RL for future works. As shown in Table 4 and Figure 4, our reasoning model achieves significantly better performance than Phi4-mini-Reasoning (which has a final RL training stage) on AIME24/25 [Art], Math500 [HBK+21], and GPQA Diamond [RHS+23], while employing a substantially more efficient architecture, achieving up to $10\times$ higher throughput in long-generation scenarios and $4.9\times$ speedup in long-context processing. In Figure 4, we evaluate the throughput of our vLLM implementation[2] using random model weights to eliminate the influence of potentially shorter generation lengths on speed measurements, ensuring a fair comparison across different architectures. The same hyperparameter configurations as Phi4-mini-Flash are applied for the YOCO and SambaY based baseline architectures. We observe that SambaY achieves the best throughput in both long-context and long-generation settings across various numbers of concurrent clients, highlighting the significant practical efficiency gains enabled by our GMU modules. Notably, our Differential Attention implementation relies on a naive four-pass of the FlashAttention [Dao23] operator for vLLM compatibility, rather than the optimized custom kernel proposed in the original paper, leaving significant room for further speed

---

[2]We customize the official vLLM framework with the version 0.7.3 to support our Phi4-mini-Flash architecture.

optimization of Phi4-mini-Flash-Reasoning to catch up the efficiency of SambaY. More evaluation details and case studies on our model's general reasoning ability are provided in Appendix G.

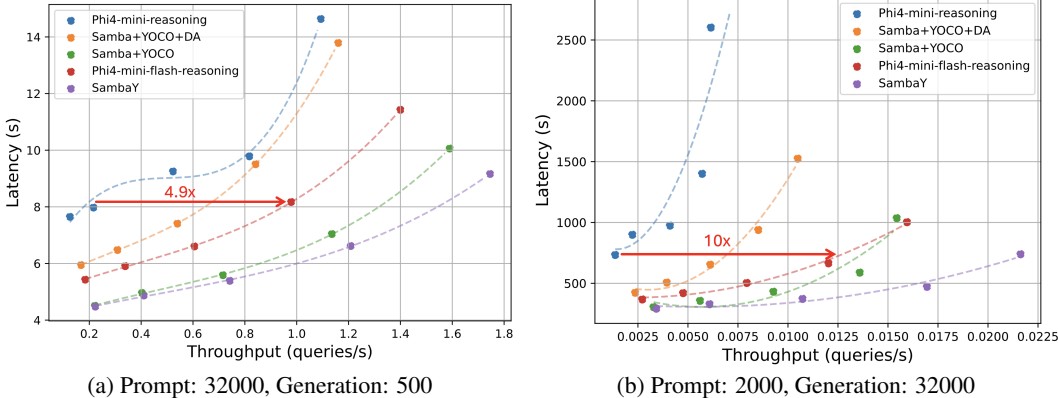

(a) Prompt: 32000, Generation: 500

(b) Prompt: 2000, Generation: 32000

Figure 4: Throughput and latency of text generation with various architectures under the vLLM inference framework (using one A100-80GB GPU and no Tensor Parallelism). A normal distribution with 30% variance was applied to prompt and generation lengths with averages of 32000/2000 and 500/32000 respectively, following the setting in [HTW+24].

## 4 Ablation Study

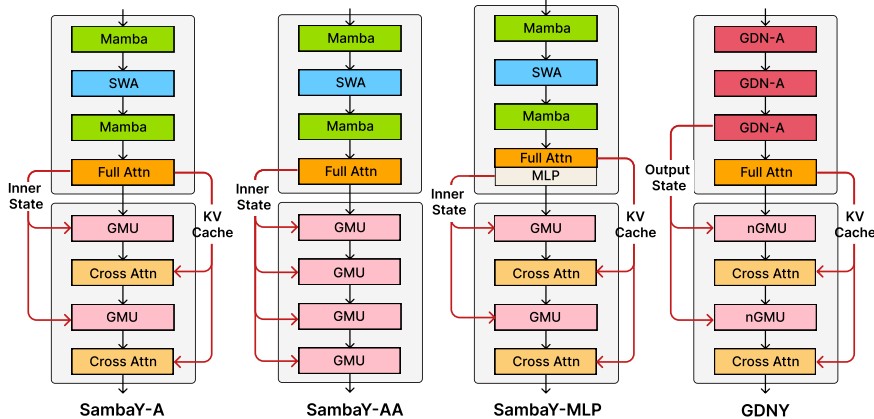

Figure 5: Major architectural variants explored in this section. For GDNY, we use Gated DeltaNet [YKH25] with normalization *after* output gate (GDN-A) for self-decoder, and apply normalized GMU (nGMU) in cross-decoder.

As illustrated in Figure 5, we systematically study the design choices in our decoder-hybrid-decoder architecture through the following architectural modifications of SambaY:

- SambaY-2 or S-GDNY substitutes Mamba layers with Mamba-2 or GDN-A (as detailed in Appendix B) respectively in the self-decoder; MambaY/MambaY-2/GDNY employs only Mamba/Mamba-2/GDN-A respectively in the self-decoder except the full attention layer. We find it is crucial to also use nGMU for Mamba-2/GDN-A based models to achieve strong long context performance, with ablation studies in Appendix H.

- SambaY-A applies GMUs to gate intermediate representations from the last full attention layer in the self-decoder rather than from Mamba.

- SambaY-AA entirely removes cross-attention in the cross-decoder and instead uses GMU to gate the intermediate representations from the middle full attention layer.

- SambaY-MLP uses GMUs to gate the intermediate representation from the linear projection branch of the SwiGLU right after the full attention layer.

All ablation models are trained with 1.0B parameters on the ProLong-64K dataset with 40B tokens and a maximum of 32K sequence length with variable length training, using a SWA size of 128 as in

the scaling experiments. We leverage $\mu$P++ with depth $d = 16$ and construct iso-parameter equations to maintain parameter count equivalence across all variants, with more details in Appendix D. We aim to answer the following research questions given the ablation results in Table 5.

Table 5: Downstream evaluation on Phonebook 32K (PB-32k), language modeling and common-sense reasoning tasks in zero-shot for 1B-parameter models with a sliding window size of 128. We measure the training speed in MTPS (Million Tokens Per Second) with 64 A100-80GB GPUs. The average accuracy excludes PB-32K due to its relatively high variability, with a standard deviation of around 5%. Best results in bold, second best underlined.

| Model | Speed mtps ↑ | Wiki. ppl ↓ | PB-32K acc ↑ | LMB. acc ↑ | ARC-c acc_n ↑ | ARC-e acc ↑ | Hella. acc_n ↑ | PIQA acc ↑ | Wino. acc ↑ | Avg. acc ↑ |
|---|---|---|---|---|---|---|---|---|---|---|
| SambaY | 1.10 | 16.89 | 78.13 | 50.22 | 28.58 | 59.18 | **49.07** | 70.84 | **55.09** | 52.16 |
| MambaY | 0.94 | 17.29 | 12.50 | 50.24 | 28.84 | 59.64 | 48.27 | 71.44 | 52.80 | 51.87 |
| SambaY-2 | **1.43** | 17.17 | 40.63 | 48.96 | 28.84 | 59.18 | 48.01 | 70.18 | 50.83 | 51.00 |
| MambaY-2 | 1.38 | 18.63 | 50.78 | 49.58 | 28.24 | 58.75 | 48.29 | 70.13 | 51.07 | 51.01 |
| S-GDNY | 1.34 | **16.78** | 83.59 | **50.94** | 29.61 | 58.96 | 48.93 | 71.55 | 51.85 | 51.97 |
| GDNY | 1.22 | 16.92 | **89.84** | 50.38 | 28.84 | 60.61 | 48.01 | 71.27 | 51.38 | 51.75 |
| SambaY-A | 1.11 | 18.12 | 58.59 | 49.85 | 30.29 | 59.60 | 48.41 | 71.33 | 54.06 | 52.26 |
| SambaY-AA | 1.25 | 17.03 | 46.88 | 49.93 | 28.50 | 59.05 | 48.69 | **72.25** | 53.91 | 52.06 |
| SambaY-MLP | 1.15 | 18.70 | 64.84 | 50.16 | **30.38** | **60.69** | 48.46 | 71.44 | 54.78 | **52.65** |

**Alternative architectures for self-decoder in SambaY?**   As shown in Table 5, while SambaY performs well on the PB-32K benchmark, replacing its Mamba layers with Mamba-2 leads to a significant drop in accuracy. This may be attributed to Mamba-2's coarse, scalar-valued forget gates, which can reduce the self-decoder's capacity to encode fine-grained positional information. The weaker PB-32K performance of MambaY compared to SambaY underscores the importance of local retrieval ability provided by SWA; recency bias alone appears insufficient for the self-decoder to support the cross-decoder in completing complex retrieval tasks. While GDN-based models achieve impressive PB-32K accuracy due to their enhanced retrieval capabilities with delta update rules, interleaving GDN with short-range SWA notably accelerates training without significantly degrading performance on either short or long-context tasks.

**Is GMU effective for other memories beyond SSMs?**   We examine SambaY-A and SambaY-AA, which gate attention inner representations, and SambaY-MLP, which gates MLP intermediate representations. As shown in Table 5, these variants achieve respectable performance on downstream tasks, with SambaY-MLP even surpassing the original SambaY on average accuracy for short-context tasks. However, for the long-context task, PB-32K, we observe a clear hierarchy: SambaY > SambaY-MLP > SambaY-A > SambaY-AA. This pattern indicates that GMUs remain effective with alternative memory sources, but their performance on retrieval tasks depends significantly on the memory source's inherent characteristics. Gating attention/MLP representations performs worse than the original SambaY on Phonebook because they lack the recency bias that SSMs naturally provide, which is beneficial for encoding contiguous local information. SambaY-AA, which completely removes cross-attention, shows significant degradation, highlighting the importance of having a sufficient number of cross-attention layers for the successful retrievals from a large pool of multiple key-value pairs.

## 5   Conclusion

In this work, we introduce the Gated Memory Unit (GMU), a simple yet effective mechanism for efficient memory sharing across layers in sequence models. Replacing expansive cross attention layers with GMUs, we propose SambaY, a decoder-hybrid-decoder architecture with Samba as the self-decoder, which achieves significant improvements in both computation efficiency and long-context performance. Our extensive scaling experiments demonstrated that SambaY exhibits a lower irreducible loss compared to strong baselines when fitted with power laws against training FLOPs, indicating superior scaling properties with increasing computational resources. Our largest model, Phi4-mini-Flash-Reasoning, outperforms Phi4-mini-Reasoning on challenging reasoning benchmarks while delivering substantially higher decoding throughput on long-context generations. Given that our architecture still retains a full attention layer with linear decoding complexity, future work could explore dynamic sparse attention mechanisms to further improve efficiency on extremely long sequence generation, particularly in agentic application scenarios.

# Acknowledgement

We want to thank Yutao Sun, Li Dong, Songlin Yang and Yang Liu for helpful discussions and insights. We also want to thank Yi Zhu for an early version of the vLLM implementation of YOCO.

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

# A  Background

YOCO (You Only Cache Once) [SDZ$^+$24] is an inference-efficient decoder–decoder architecture with linear pre-filling complexity. It comprises a self-decoder, formed by the first half of the layers, which employ token mixers with linear computational complexity and include a final full-attention layer that generates Key-Value (KV) caches during pre-filling. And the second half of the layers forms the cross-decoder, which are cross-attention layers that attend to the KV caches produced by the self-decoder's last full attention layer. Specifically, for an input sequence of hidden states $X_{mem} \in \mathbb{R}^{n \times d_m}$ of the final full-attention layer in the self-decoder, it pre-computes and caches a sequence of KV pairs during pre-filling:

$$K_c = X_{mem} W_K, \quad V_c = X_{mem} W_V,$$

where $K_c, V_c \in \mathbb{R}^{n \times d_{kv}}$ are the cached matrices and $W_K, W_V$ are weight matrices. Subsequently, at the decoding stage, every cross-attention layer $l$ in the cross-decoder reuses this single set of KV cache. Given the input hidden state $X_{cross}^{(l-1)}$ to that layer, the cross-attention output is calculated by generating a new query, $Q_{cross}^{(l)}$, and attending to the shared cache:

$$Q_{cross}^{(l)} = X_{cross}^{(l-1)} W_Q^{(l)},$$

$$H^{(l)} = \text{softmax}\left(\frac{Q_{cross}^{(l)} K_c^T}{\sqrt{d_{kv}}}\right) V_c.$$

During pre-filling, this approach (1) entirely avoids the computation of full attention and (2) requires inference through only the first half of the layers, substantially reducing the computational cost of processing user prompts for both short and long contexts. Our SambaY architecture further improves its efficiency by modifying the cross-decoder: we replace half of its memory I/O-expensive cross-attention layers with lightweight Gated Memory Units (GMUs), thereby enhancing decoding efficiency during response generation.

# B  Additional Theoretical Analysis

**Normalization placement in linear attention.**  In linear attention architectures [QHS$^+$22, SDH$^+$23, YWS$^+$23], including Gated DeltaNet (GDN) [YKH25], a normalization operator is applied immediately after token mixing to stabilize training at layer $l'$:

$$M^{(l')} = \text{Norm}\big(A^{(l')} V^{(l')}\big), \qquad \mathbf{y}^{(l')} = \big(M^{(l')} \odot G^{(l')}\big) W_2^{(l')}, \qquad G^{(l')} = \sigma(W_1^{(l')} X^{(l')}),$$

where $\text{Norm}$ is typically RMSNorm [ZS19] and $X^{(l')}$ is the layer input and $Y^{(l')}$ is the layer output. Placing $\text{Norm}$ *before* the output gating, however, breaks the associativity between the gating matrix $G^{(l')}$ and the token-mixing operator $A^{(l')}$, so the Gated Memory Unit (GMU) can no longer directly re-weight the token mixing based on the current layer input with $X^{(l)}$. To resolve this issue, we propose to postpone the $\text{Norm}$ *after* output gating for GDN (denoted as GDN-A), following the design of Mamba-2 [DG24]. Concretely, for layer $l'$ we instead compute

$$M^{(l')} = A^{(l')} V^{(l')}, \qquad Y^{(l')} = \text{Norm}\big(M^{(l')} \odot G^{(l')}\big) W_2^{(l')},$$

and employ the normalized GMU (nGMU) at layer $l > l'$ in the cross-decoder to maintain training stability while allowing the gate to modulate $A^{(l')}$ directly, *i.e.*

$$Y^{(l)} = \text{Norm}\big(M^{(l')} \odot G^{(l)}\big) W_2^{(l)}.$$

This simple reordering preserves associative re-weighting and, as demonstrated empirically in Appendix H, substantially improves the long context performance when compared to the original normalization before gating design in GDN.

## C Additional Aspect Ratio Calculations

When solving for the aspect ratio through iso-parametric equations, we rounded it up to an even integer to guarantee the activation of Tensor Cores[3]. Based on the Samba+YOCO architecture, we can derive the iso-parametric equation through calculating the number of non-embedding parameters as follows,

$$N_{\text{attn}}(d) = 2.5dw \cdot w_{\text{attn}}/4 + 2dw \cdot w_{\text{attn}}/2, \quad N_{\text{mamba}}(d) = 6dw^2/4,$$

$$N(d) = N_{\text{attn}}(d) + N_{\text{mamba}}(d) + N_{\text{mlp}}(d) = 208\alpha d^3 + 13.5\alpha^2 d^3 = 237568d^3.$$

Solving for $\alpha$, we get $\alpha_2 \approx 126$. For the SambaY+DA architecture, the aspect ratio is not changed because the number of extra parameters introduced by DA is negligible. For MambaY, we have

$$N_{\text{attn}}(d) = 2dw \cdot w_{\text{attn}}/4, \quad N_{\text{mamba}}(d) = 6dw^2/2, \quad N_{\text{gmu}}(d) = 4dw^2/4,$$

$$N(d) = N_{\text{attn}}(d) + N_{\text{mamba}}(d) + N_{\text{mlp}}(d) + N_{\text{gmu}}(d) = 64\alpha d^3 + 16\alpha^2 d^3 = 237568d^3.$$

Solving for $\alpha$, we get $\alpha_3 \approx 120$. For SambaY-MLP, we have

$$N_{\text{attn}}(d) = 2.5dw \cdot w_{\text{attn}}/4 + 2dw \cdot w_{\text{attn}}/4, \quad N_{\text{mamba}}(d) = 6dw^2/4, \quad N_{\text{gmu}}(d) = 8dw^2/4,$$

$$N(d) = N_{\text{attn}}(d) + N_{\text{mamba}}(d) + N_{\text{mlp}}(d) + N_{\text{gmu}}(d) = 144\alpha d^3 + 15.5\alpha^2 d^3 = 237568d^3.$$

Solving for $\alpha$, we get $\alpha_4 \approx 120$. For SambaY-Attn, we have

$$N_{\text{attn}}(d) = 2.5dw \cdot w_{\text{attn}}/4 + 2dw \cdot w_{\text{attn}}/4, \quad N_{\text{mamba}}(d) = 6dw^2/4, \quad N_{\text{gmu}}(d) = 2dw \cdot w_{\text{attn}}/4,$$

$$N(d) = N_{\text{attn}}(d) + N_{\text{mamba}}(d) + N_{\text{mlp}}(d) + N_{\text{gmu}}(d) = 208\alpha d^3 + 13.5\alpha^2 d^3 = 237568d^3.$$

Solving for $\alpha$, we get $\alpha_5 \approx 126$, which is the same as Samba+YOCO. For SambaY-Attn-All, we similarly have

$$N_{\text{attn}}(d) = 2.5dw \cdot w_{\text{attn}}/4, \quad N_{\text{mamba}}(d) = 6dw^2/4, \quad N_{\text{gmu}}(d) = 2dw \cdot w_{\text{attn}}/2,$$

$$N(d) = N_{\text{attn}}(d) + N_{\text{mamba}}(d) + N_{\text{mlp}}(d) + N_{\text{gmu}}(d) = 208\alpha d^3 + 13.5\alpha^2 d^3 = 237568d^3.$$

Solving for $\alpha$, we get $\alpha_6 \approx 126$. For the GDNY architecture, we use a fixed head dimension of 256 and SwiGLU output gating with the Gated DeltaNet layers. The query-key projections with a 0.75 expansion ratio are used, while the value, gating, and output projections are using a 1.5 expansion ratio with respect to the model width. We also allow negative eigenvalues for improved expressiveness of the transition matrices [GSF+25]. Specifically, we have

$$N_{\text{attn}}(d) = 2dw \cdot w_{\text{attn}}/4, \quad N_{\text{GDN}}(d) = 6dw^2/2, \quad N_{\text{gmu}}(d) = 3dw^2/4,$$

$$N(d) = N_{\text{attn}}(d) + N_{\text{GDN}}(d) + N_{\text{mlp}}(d) + N_{\text{gmu}}(d) = 64\alpha d^3 + 15.75\alpha^2 d^3 = 237568d^3.$$

Solving for $\alpha$, we get $\alpha_7 \approx 120$. As in SambaY, we can similarly solve for S-GDNY to get $\alpha_8 \approx 126$ and for SWA+YOCO to get $\alpha_9 \approx 130$.

## D Implementation Details

**Details on scaling comparisons.** Except for the learning rate, we fix other hyper-parameters of the AdamW optimizer with $\beta_1 = 0.9, \beta_2 = 0.95, \epsilon = 10^{-8}$ and a weight decay of 0.1. A learning rate schedule is applied with 1B warm-up tokens linearly increasing to the peak learning rate $\eta$, followed by a linear decay to zero. We use LeCun uniform initialization (*i.e.* PyTorch default initialization) [LBOM12] for the weight matrices following [GD23] and [RLL+25], and tie the input and output embedding matrices which are initialized from the normal distribution $\mathcal{N}(0, 0.02^2)$. The attention logits scaler is set to $1/\sqrt{d_{kv}}$, where $d_{kv}$ is the head dimension. We summarize the key differences between $\mu$P, $\mu$P++ and Standard Parameterization (SP) in Table 6, with additional details as follows. For $\mu$P++, we scale the output logits and the learning rate of matrix-like parameters proportional to $1/w$. The output of each layer is divided by $\sqrt{2d}$ following Depth-$\mu$P.

For SP, we don't apply any $\mu$P++ scaling laws, and since LeCun initialization already scales its initialization variance with respect to $1/d_{in}$ as the same as proposed in $\mu$P, where $d_{in}$ is the fan-in

---

[3] https://developer.nvidia.com/blog/optimizing-gpu-performance-tensor-cores/

Table 6: Key differences between $\mu$P, $\mu$P++ and Standard Parameterization (SP). *LR mult.* denotes the per-parameter multiplier applied on top of the global learning-rate ($\eta$), *Res. mult.* is the multiplier applied to the output of residual branches and *WD* denotes the weight decay. For $\mu$P++, $\eta \propto 1/\sqrt{d}$ and zero weight decay is also applied to other scalar or vector-like parameters such as RMSNorm weights. In this work, $\sigma = 10^{-4}$ for untied embedding and $\sigma = 0.02$ for tied embedding, and in both cases $\tau = 0.02$ and $\beta = 1$. "fan_in" means the input dimension of weight matrices.

| Parameter | Scheme | LR mult. | Initialization | Res. mult. | Weight mult. | WD |
|---|---|---|---|---|---|---|
| Embedding | SP | $\propto 1$ | $\mathcal{N}(0, \sigma^2)$ | — | $\propto 1$ | $\propto 1$ |
| | $\mu$P | $\propto 1$ | $\mathcal{N}(0, \sigma^2)$ | — | $\propto 1$ | $\propto 1$ |
| | $\mu$P++ | $\propto 1$ | $\mathcal{N}(0, \sigma^2)$ | — | $\propto 1$ | 0 |
| Unembedding | SP | $\propto 1$ | 0 or tied | — | $\propto 1$ | $\propto 1$ |
| | $\mu$P | $\propto 1$ | 0 or tied | — | $\propto 1/w$ | $\propto 1$ |
| | $\mu$P++ | $\propto 1$ | 0 or tied | — | $\propto 1/w$ | 0 |
| Hidden Weights | SP | $\propto 1$ | $\mathcal{N}(0, \tau^2)$ | 1 | $\propto 1$ | $\propto 1$ |
| | $\mu$P | $\propto 1/w$ | $\mathcal{U}(\frac{-\beta}{\sqrt{\text{fan\_in}}}, \frac{\beta}{\sqrt{\text{fan\_in}}})$ | 1 | $\propto 1$ | $\propto 1$ |
| | $\mu$P++ | $\propto 1/w$ | $\mathcal{U}(\frac{-\beta}{\sqrt{\text{fan\_in}}}, \frac{\beta}{\sqrt{\text{fan\_in}}})$ | $1/\sqrt{2d}$ | $\propto 1$ | $\propto 1$ |

dimension of the weight matrix, we instead use normal initialization with a standard deviation of 0.02 for weight matrices to rule out this confounding effect. We divide the initialization standard deviation of the output projection of each layer by $\sqrt{2d}$, following [RWC+19, GD23, RLL+25]. The detailed architecture and optimization setups for each of the scales are shown in Table 7. Following [GD23, YWZ+24, RLL+25, YKH25], our downstream evaluations are conducted on the following benchmarks: Wikitext [MXBS16], LAMBADA (LMB) [PKL+16], Arc-Easy/Challenge (ARC-e/ARC-c) [CCE+18], HellaSwag (Hella.) [ZHB+19], WinoGrande (Wino.) [SBBC21] and PIQA [BZB+20], where we measure character normalized accuracy (acc_n) for Arc-Challenge and HellaSwag.

**More details on architecture and large-scale pre-training.** We provide a comprehensive summary of the architectures explored in this work, along with the large-scale pre-training setup, in Table 7. In our architectures, Differential Attention uses a depth-dependent initialization factor, $\lambda_{\text{init}} = 0.8 - 0.6 \exp(-0.3 \times l)$, where $l$ is the depth index. For each attention head, it employs two sets of learnable parameters, $(\lambda_{q1}, \lambda_{k1})$ and $(\lambda_{q2}, \lambda_{k2})$, each of dimension equal to the head dimension and initialized with a normal distribution of zero mean and 0.1 standard deviation. RMSNorm [ZS19] with learnable element-wise affine parameters is adopted for attention output normalization. For each of the intermediate layers, LayerNorm [BKH16] is used with Pre-LN [XYH+20] for Phi4-mini-Flash architecture.

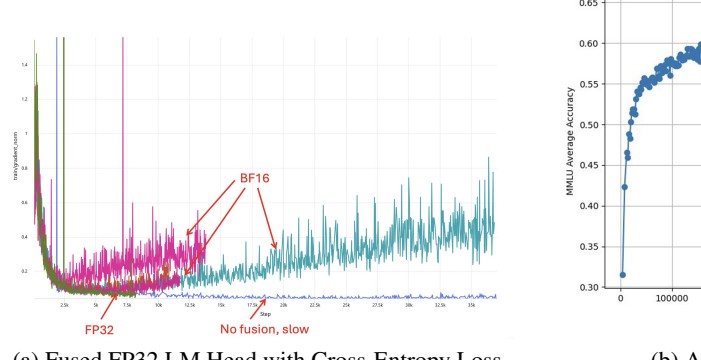
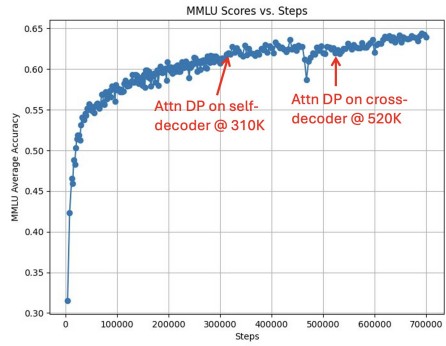

(a) Fused FP32 LM Head with Cross-Entropy Loss  (b) Adding Attention Dropout

Figure 6: Adopted tricks for mitigating the large-scale pre-training instability of Phi4-mini-Flash.

**Mitigation of instability in large scale pre-training.** During the pre-training stage of Phi4-mini-Flash, we meet severe loss divergence, which is mitigated with the following two tricks: (1) we

up-cast the weight and the input to FP32 during chunk-wise matrix multiplication for our fused linear cross-entropy loss kernel which is modified from the Liger-Kernel[4]. As shown in Figure 6a, with the FP32 up-casting, the gradient norm during the training process can be stabilized to closely track the naive no-fusion baseline, compared to the normal BF16 matrix multiplications with up-shooting trends that will finally blow up the training loss. Without fusion, the training speed is substantially slower because of our large 200K vocabulary size. (2) As shown in Figure 6b, we add 0.05 attention dropout at 310K steps for self-decoder and at 520K steps for cross-decoder, which are the last checkpoint steps before the loss divergences happen. As we can see from the figure, adding attention dropout doesn't harm downstream performance on MMLU [HBB$^+$21], but can successfully stabilize the whole training process until the end.

Table 7: Model and training configurations for the architectures explored in this work. TransformerLS adopts the same architecture as Transformer++, with Sliding Window Attention (SWA) applied to all attention layers except every fourth layer, which uses full attention. MLP Size denotes the intermediate dimension of the MLP, *i.e.*, the input dimension of the output projection. Phi4-mini and Phi4-mini-Flash are trained with a batch size of 8M tokens, using a linear learning rate schedule with 3,000 warm-up steps. We allow the intermediate dimension of attention layers to be larger than the model width, so that the head dimension can be a power of 2. Variants enhanced with Differential Attention adopt the same architectural configurations as their respective baselines. All models use tied embeddings. The 3.8B-parameter SambaY and Samba+YOCO models are randomly initialized for benchmarking under the vLLM inference framework. Except for 3.8B-parameter models which use a vocabulary of 200K tokens, we apply Llama-2 [TMS$^+$23] tokenizer with a 32K vocabulary for all other models.

| Architecture | Depth $d$ | Model Width | Query Heads | KV Heads | Head Dim | MLP Size | Non-Embed Params (M) | Params (M) | Learning Rate | Training Tokens (B) |
|---|---|---|---|---|---|---|---|---|---|---|
| Transformer++ | 8 | 1024 | 8 | 2 | 128 | 4096 | 121.6 | 154.4 | 5.66e-04 | 12.5 |
| | 12 | 1536 | 12 | 3 | 128 | 6144 | 410.5 | 459.7 | 4.62e-04 | 42.2 |
| | 16 | 2048 | 16 | 4 | 128 | 8192 | 973.1 | 1038.6 | 4.00e-04 | 100.0 |
| | 20 | 2560 | 20 | 5 | 128 | 10240 | 1900.5 | 1982.5 | 3.58e-04 | 195.3 |
| | 24 | 3072 | 24 | 6 | 128 | 12288 | 3284.1 | 3382.4 | 3.27e-04 | 337.5 |
| SambaY | 8 | 992 | 8 | 2 | 128 | 3968 | 123.3 | 155.0 | 5.66e-04 | 12.7 |
| | 12 | 1488 | 12 | 3 | 128 | 5952 | 416.1 | 463.7 | 4.62e-04 | 42.8 |
| | 16 | 1984 | 16 | 4 | 128 | 7936 | 986.3 | 1049.8 | 4.00e-04 | 101.4 |
| | 20 | 2480 | 20 | 5 | 128 | 9920 | 1926.5 | 2005.8 | 3.58e-04 | 198.0 |
| | 24 | 2976 | 24 | 6 | 128 | 11904 | 3328.9 | 3424.2 | 3.27e-04 | 342.1 |
| Samba+YOCO | 8 | 1008 | 8 | 2 | 128 | 4032 | 123.2 | 155.4 | 5.66e-04 | 12.7 |
| | 12 | 1512 | 12 | 3 | 128 | 6048 | 415.6 | 464.0 | 4.62e-04 | 42.7 |
| | 16 | 2016 | 16 | 4 | 128 | 8064 | 985.2 | 1049.7 | 4.00e-04 | 101.2 |
| | 20 | 2520 | 20 | 5 | 128 | 10080 | 1924.3 | 2004.9 | 3.58e-04 | 197.8 |
| | 24 | 3024 | 24 | 6 | 128 | 12096 | 3325.1 | 3421.9 | 3.27e-04 | 341.7 |
| SWA+YOCO | 16 | 2080 | 16 | 4 | 128 | 8320 | 984.0 | 1050.6 | 4.00e-04 | 40.0 |
| MambaY | 16 | 1920 | 16 | 4 | 128 | 7680 | 975.2 | 1036.6 | 4.00e-04 | 40.0 |
| GDNY | 16 | 1920 | 16 | 4 | 128 | 7680 | 960.4 | 1021.9 | 4.00e-04 | 40.0 |
| S-GDNY | 16 | 2016 | 16 | 4 | 128 | 7680 | 1001.0 | 1065.5 | 4.00e-04 | 40.0 |
| MambaY-2 | 16 | 1920 | 16 | 4 | 128 | 7680 | 975.2 | 1036.6 | 4.00e-04 | 40.0 |
| SambaY-2 | 16 | 1984 | 16 | 4 | 128 | 7936 | 986.3 | 1049.8 | 4.00e-04 | 40.0 |
| SambaY-A | 16 | 2016 | 16 | 4 | 128 | 8064 | 985.2 | 1049.7 | 4.00e-04 | 40.0 |
| SambaY-AA | 16 | 2016 | 16 | 4 | 128 | 8064 | 985.2 | 1049.7 | 4.00e-04 | 40.0 |
| SambaY-MLP | 16 | 1920 | 16 | 4 | 128 | 7680 | 985.0 | 1046.4 | 4.00e-04 | 40.0 |
| Phi4-mini | 32 | 3072 | 24 | 8 | 128 | 8192 | 3221.2 | 3835.8 | 5.00e-04 | 5000 |
| Pih4-mini-Flash | 32 | 2560 | 40 | 20 | 64 | 10240 | 3329.2 | 3841.4 | 5.00e-04 | 5000 |
| SambaY | 32 | 2560 | 40 | 20 | 64 | 10240 | 3329.2 | 3841.4 | - | - |
| Samba+YOCO | 32 | 2560 | 40 | 20 | 64 | 10240 | 3224.4 | 3736.5 | - | - |

# E  Ablation Study on Hyper-parameter Scaling Laws

We conduct a comprehensive ablation study of our $\mu$P++ scaling laws to validate their scaling behavior. All experiments are performed using Transformer++ trained with a 4K sequence length

---
[4] https://github.com/linkedin/Liger-Kernel

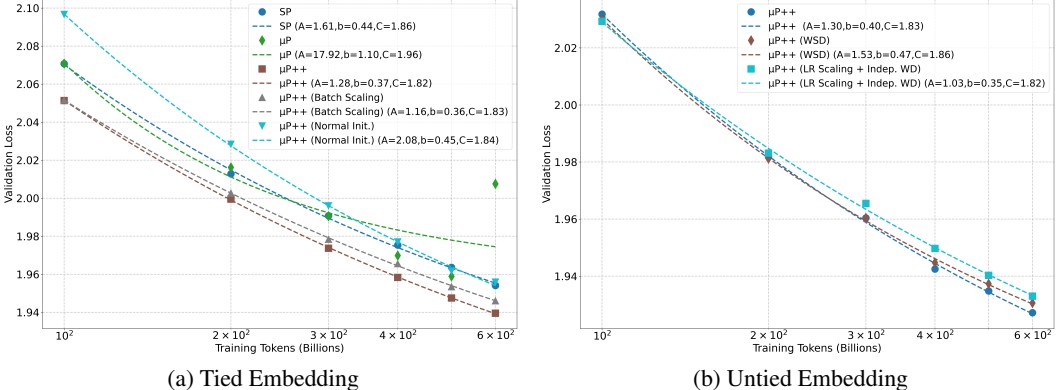

|                | (a) Tied Embedding | (b) Untied Embedding |
|----------------|--------------------|----------------------|

Figure 7: Validation Loss v.s. Training Tokens on the SlimPajama dataset for Transformer++ trained with tied (left) or untied (right) embedding layers. For the training on 600B tokens with $\mu$P, the model encountered NaN losses after 204K gradient update steps. We report the last valid validation loss prior to divergence as its final performance.

on the SlimPajama dataset. To ensure that the linear learning rate schedule fully decays to zero, we train six models at different training token budgets: $\{100\text{B}, 200\text{B}, \ldots, 600\text{B}\}$ for each of the scaling curves. We examine the scaling performance under both tied and untied embedding setups. For the untied setting, we follow RWKV [PAA$^+$23] by applying normal initialization with zero mean and a standard deviation of $10^{-4}$. The unembedding layer is initialized to zero, following the zero-out trick proposed in $\mu$P [YHB$^+$22]. As shown in Figure 7a, we observe that the original $\mu$P setup (which uses LeCun initialization and does not include Depth-$\mu$P or weight decay modifications as in $\mu$P++) can lead to severe training instability when scaling to 600B tokens. Since we observe increasing gradient norms with large spikes for vector-like parameters shortly before the model diverges, this highlights the importance of the $\mu$P++ strategy of applying zero weight decay to vector-like parameters to enhance training stability at large scales. We also explore batch size scaling with respect to training token size, following [SWW$^+$24, LZH$^+$25], *i.e.*

$$B = B_0 \sqrt{\frac{T}{T_0}}.$$

As in Figure 7a, $\mu$P++ (Batch Scaling) shows both worse learning efficiency and irreducible loss than $\mu$P++. Generally, we think the batch size mainly affects parallelism and the computation efficiency as long as the batch size is not too large, and the worse scaling behavior can be because (1) when scaling up, the batch size can surpass the critical batch size [MKAT18], which leads to worse model performance, (2) other optimizer hyper-parameters are not adjusted accordingly with batch size as in [MLPA22] and we leave it for future works to study the large batch size training with $\mu$P++. We also try using Normal Initialization with 0.02 standard deviation for the weight matrices, and scale the variance with respect to $1/d$. However, $\mu$P++ (Normal Init.) shows worse scaling than $\mu$P++, indicating that it is better to adjust the initialization multipliers based on each matrix' dimension as adopted by LeCun initialization, rather than a global factor related to model width. We explore integrating the empirical scaling law of the learning rate $\eta$ scaling with respect to training tokens $T$ [BBC$^+$25] to $\mu$P++, *i.e.*,

$$\eta = \eta_0 \sqrt{\frac{Bd_0}{B_0 d}} \left(\frac{T_0}{T}\right)^{\frac{1}{3}},$$

and adjust weight decay to maintain the same regularization effect across different training tokens with the setup of Independent Weight Decay [WLX$^+$24], *i.e.*,

$$\lambda = \lambda_0 \frac{\eta_0}{\eta},$$

where $\lambda$ is the weight decay in AdamW [LH18] and $\lambda_0 = 0.1$. We denote this scaling law as $\mu$P++ (LR scaling + Indep. WD). As in Figure 7b, while the irreducible loss is comparable, we observe a worse learning efficiency with smaller $b$ compared to $\mu$P++. We think that future work is needed to

have an empirical study of the learning rate scaling with respect to dataset size under $\mu$P++, instead of transferring the empirical law directly to our theoretical laws. We also explore using the WSD [HTH+24] learning rate scheduler for $\mu$P++, where we set the final decay period to be $2/7$ of the total period following [DA24b]. Unfortunately, it depicts worse scaling behavior than $\mu$P++ with a linear learning rate schedule, as shown in Figure 7b. Interestingly, when comparing the performance of $\mu$P++ with tied versus untied embeddings, we observe that $\mu$P++ with untied embeddings achieves a significantly lower validation loss with 100B training tokens, but its irreducible loss remains comparable to that of tied embeddings. This suggests that the additional parameters from untied embeddings primarily accelerate training convergence without improving the final model performance if a sufficient amount of data is given.

# F    Additional Long-Context Retrieval Experiments

**Long-context extrapolation with NoPE.**    In Table 8, we directly measure the retrieval accuracy at 32K, 64K and 128K context length on the Phonebook benchmark for 1B parameter models trained with 32K sequence length in Section 3.2. We can see SambaY and its variants with NoPE can extrapolate their retrieval ability by $2\times$ in zero-shot, while RoPE-based models (Transformer++ and TransformerLS) have a substantial drop beyond 32K. We leave the explanations of why NoPE can enable limited extrapolations on retrieval tasks as an interesting future work.

Table 8: Long-context extrapolation accuracy (with standard deviations) on the Phonebook benchmark. The models are trained on the ProLong-64K dataset with 32K sequence length and 1B parameters.

| Model | SWA Size | 32K Acc. (%) | 64K Acc. (%) | 128K Acc. (%) |
|---|---|---|---|---|
| Transformer++ | – | $60.94 \pm 10.00$ | $0.00 \pm 0.00$ | $0.00 \pm 0.00$ |
| TransformerLS | 256 | $60.16 \pm 7.12$ | $17.19 \pm 5.63$ | $0.78 \pm 1.35$ |
| Samba+YOCO | 1024 | $82.81 \pm 6.44$ | $67.97 \pm 11.13$ | $20.31 \pm 8.41$ |
| SambaY | 256 | $92.19 \pm 1.56$ | $96.09 \pm 2.59$ | $0.00 \pm 0.00$ |
| SambaY+DA | 512 | $96.09 \pm 1.35$ | $84.38 \pm 3.12$ | $5.47 \pm 2.59$ |

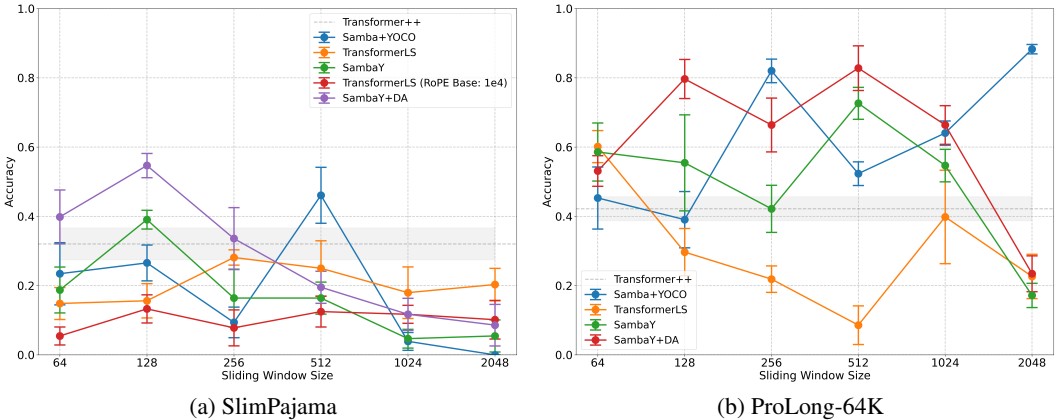

(a) SlimPajama                              (b) ProLong-64K

Figure 8: Accuracy (with error bars) v.s. Sliding Window Size on Phonebook with 32K evaluation length using 40B training tokens from SlimPajama (left) or ProLong-64K (right). As an ablation to Figure 3, variable-length training is not applied for both setting.

**Ablation on training data and methodologies.**    Figure 8 illustrates how different model architectures perform on the Phonebook long-context task as the sliding window size increases, using either SlimPajama or ProLong-64K for pre-training with 32K sequence length and without variable-length training. Specifically, we concatenate the data samples with EOS tokens as separation to form 32K length training sequences. On SlimPajama, overall accuracy is modest, with SambaY+DA showing some initial promise at smaller window sizes (peaking at 128) before declining, while Samba+YOCO performs best at a moderate window size of 512. Transformer-based models generally struggle to achieve competitive accuracy across window sizes. Notably, reducing RoPE base from 640K to

10k for TransformerLS significantly harms the performance across window sizes. Switching to the ProLong-64K dataset leads to a notable performance boost across all architectures compared to SlimPajama, even without variable-length training. We can observe that SSM-based models enjoy larger boosts on accuracies than Transformer++. This indicates that SSM-based models can learn to switch contexts between different data samples within the packed sequences more easily than pure attention models. Notably, SambaY+DA achieves competitive accuracy using a smaller sliding window (512), matching the performance of Samba+YOCO at larger window sizes. While Samba+YOCO continues to benefit from increasing window sizes, reaching peak accuracy at 2048, SambaY+DA demonstrates greater efficiency by achieving strong results with a smaller sliding window size. Given that variable-length training on ProLong-64K generally yields better results as in Figure 3, these fixed-length training results indicate that while ProLong-64K benefits long-context performance, the full potential, especially for pure attention models that are sensitive to sliding window size (e.g. TransformerLS), can be further unlocked by training methodologies that explicitly account for varying sequence lengths of each data sample. The different optimal sliding window sizes and performance trajectories underscore that both the pre-training dataset and the training methodology significantly influence how effectively the training context length can be utilized for long-context pre-training.

# G  Additional Details on Efficiency and Reasoning Results

Following Phi4-mini-Reasoning [XPA+25], the evaluation is conducted with a sampling temperature of 0.6, a top-p [HBD+19] value of 0.95, and a maximum sequence length of 32,768 tokens. We leverage the Math-Verify library[5] (version 0.7.0) and Lighteval[6] (version 0.10.0) to enable efficient and robust evaluation on reasoning tasks. We prepend the instruction: "Please reason step by step, and put your final answer within \\boxed{{}}." for the evaluation on AIME24/25 and MATH500 and "Please reason step by step, and put your final choice of one letter from A/B/C/D within \\boxed{{}}." for the evaluation on GPQA Diamond.

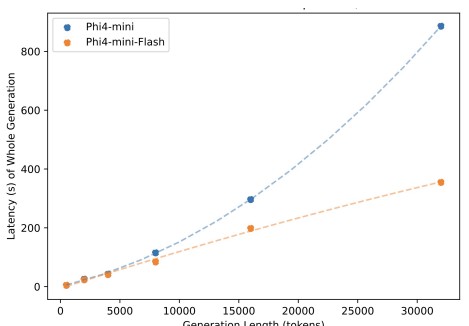

Figure 9: Generation latencies at length of 1K, 2K, 4K, 8K, 16K and 32K for a prompt length of 2000. Given a certain generation length, we measure the average latency of all the requests in all the loads of 1, 2, 4, 8, 16 concurrent requests.

The data-points of latency and throughput in Figure 4 are measured by varying the concurrency levels of the requests sent to the vLLM server, ranging from $\{1, 2, 4, 8, 16\}$. We also compare the latency of our Phi4-mini-Flash architecture with Phi4-mini across various generation lengths in Figure 9. We can see that our model can achieve faster generation speeds starting from a generation length as small as 4K. Apart from the sub-optimal Differential Attention implementation, our vLLM implementation is also sub-optimal in calculating unnecessary full attention operations with the following MLP layer for the user prompt tokens during the prefill stage, leaving additional opportunities for optimizing the efficiency results in Figure 4 and Figure 9.

We conduct case studies for Phi4-mini-Flash-Reasoning in Example 1 and Example 2 for both mathematical reasoning and daily-life reasoning. Notably, while our model is only trained on distillation data with math problems, we can see our model exhibits advanced reasoning techniques such as double checking (*e.g. "Alternatively, maybe I could have completed the square instead?", "Breaking it down:"*), reflection (*e.g. "That means there are no real solutions, right? ", "Wait, maybe I miscounted."*), and backtracking (*e.g. "But the problem might be expecting complex solutions.", "Wait, maybe I'm overcomplicating."*) in both cases. Surprisingly, in Example 2, our model will even guess the user intention (*e.g. "But wait, sometimes people might confuse the number of R's in a double letter. "*) to finally provide the correct answer.

---

[5] https://github.com/huggingface/Math-Verify
[6] https://github.com/huggingface/lighteval

## Example 1: Mathematical Reasoning (Part 1)

**Query:** solve x^2+3x+4=1

*Model's reasoning process:*

---
**Think**

Okay, let's see. I need to solve the equation x squared plus 3x plus 4 equals 1. Hmm, so first, maybe I should get all the terms on one side so that the equation equals zero. That usually makes it easier to solve quadratic equations. Let me subtract 1 from both sides. That would give me $x^2 + 3x + 4 - 1 = 0$. Simplifying that, 4 minus 1 is3, so the equation becomes $x^2 + 3x + 3 = 0$. Alright, now I have a quadratic equation in standard form, which is $ax^2 + bx + c = 0$. Here, a is 1, b is 3, and c is 3. To solve this, I can use the quadratic formula, which is x equals negative b plus or minus the square root of b squared minus four ac all over 2a. Let me write that down: x $= [-b \pm \sqrt{(b^2 - 4ac)}] / (2a)$.

Plugging in the values from my equation, b is 3, so negative b would be -3. Then, b squared is 3 squared, which is 9.Then, 4ac is 4 times 1 times 3, which is 12. So the discriminant, which is the part under the square root, would be 9 - 12. That equals -3. Wait, so the discriminant is negative. That means there are no real solutions, right? Because you can't take the square root of a negative number in the real number system.

But the problem might be expecting complex solutions. Let me check. If complex numbers are allowed, then the square root of -3 can be written as i times the square root of 3,where i is the imaginary unit. So, substituting back into the quadratic formula, we have x equals $[-3 \pm \sqrt{(-3)}] / (2*1)$. Which simplifies to $x = [-3 \pm i\sqrt{3}]/2$.

Let me double-check my steps to make sure I didn't make a mistake. Starting equation: $x^2 + 3x + 4 = 1$. Subtract 1: $x^2 + 3x + 3 = 0$. Coefficients a=1, b=3, c=3. Discriminant: $3^2 - 4*1*3 = 9 - 12 = -3$. Yep, that's right. So the solutions are complex: $(-3 \pm i\sqrt{3})/2$.

Alternatively, maybe I could have completed the square instead? Let me try that method to verify. Starting with $x^2 + 3x + 3 = 0$. Move the constant term to the other side: $x^2 + 3x = -3$. To complete the square, take half of the coefficient of x, which is 3/2, square it to get (9/4). Add that to both sides: $x^2 + 3x + 9/4 = -3 + 9/4$. The left side becomes $(x + 3/2)^2$. The right side: -3 is -12/4, so -12/4 + 9/4 is -3/4. So $(x + 3/2)^2 = -3/4$. Taking the square root of both sides gives $x + 3/2 = \pm\sqrt{(-3/4)} = \pm i\sqrt{(3/4)} = \pm i(\sqrt{3})/2$. Subtract 3/2 from both sides: $x = -3/2 \pm (i\sqrt{3})/2$. Which is the same as $(-3 \pm i\sqrt{3})/2$. So that matches the quadratic formula result.

Therefore, both methods lead to the same complex solutions. Since the original equation has a negative discriminant, there are no real solutions, only complex ones. The answer should be expressed in terms of complex numbers.

Just to be thorough, let me plug one of the solutions back into the original equation to verify. Let's take $x = [-3 + i\sqrt{3}]/2$. Plugging into $x^2 + 3x + 4$.

First, compute $x^2$: $[(-3 + i\sqrt{3})/2]^2$. Let's expand that.

$(-3 + i\sqrt{3})^2 = (-3)^2 + 2*(-3)*(i\sqrt{3}) + (i\sqrt{3})^2 = 9 - 6i\sqrt{3} + (i^2*(\sqrt{3})^2) = 9 - 6i\sqrt{3} + (-1)*3 = 9 - 6i\sqrt{3} - 3 = 6 - 6i\sqrt{3}$. Then divide by 4: $(6 - 6i\sqrt{3})/4 = (3 - 3i\sqrt{3})/2$.

Next, compute 3x: $3*[(-3 + i\sqrt{3})/2] = (-9 + 3i\sqrt{3})/2$.

Add $x^2 + 3x + 4$:

$(3 - 3i\sqrt{3})/2 + (-9 + 3i\sqrt{3})/2 + 4$. Combine the fractions: $[3 - 3i\sqrt{3} -9 + 3i\sqrt{3}]/2 + 4 = (-6)/2 + 4 = -3 + 4 = 1$. Which matches the right side of the original equation. So the solution checks out.

Similarly, if I plug in the other solution $x = [-3 - i\sqrt{3}]/2$, the process would be the same, and the imaginary parts would cancel out as well, resulting in 1.

Therefore, the solutions are correct. So the final answer is two complex numbers: $(-3 \pm i\sqrt{3})/2$. I think that's all. The key steps were moving all terms to one side, calculating the discriminant, recognizing it's negative, and proceeding with complex solutions. Both quadratic formula and completing the square worked, and verifying by substitution confirmed the answers are correct.

---

*To be continued...*

## Example 1: Mathematical Reasoning (Part 2)

**Model's Response (continued):**
To solve the equation $x^2 + 3x + 4 = 1$:
1. **Rearrange the equation** to standard quadratic form:

$$x^2 + 3x + 4 - 1 = 0 \implies x^2 + 3x + 3 = 0$$

2. **Identify coefficients**: $a = 1$, $b = 3$, $c = 3$.
3. **Calculate the discriminant**:

$$\Delta = b^2 - 4ac = 3^2 - 4(1)(3) = 9 - 12 = -3$$

Since the discriminant is negative, solutions are complex.
4. **Apply the quadratic formula**:

$$x = \frac{-b \pm \sqrt{\Delta}}{2a} = \frac{-3 \pm \sqrt{-3}}{2} = \frac{-3 \pm i\sqrt{3}}{2}$$

5. **Verify via completing the square**:

$$x^2 + 3x = -3 \implies \left(x + \frac{3}{2}\right)^2 = -\frac{3}{4}$$

Solving gives $x = -\frac{3}{2} \pm \frac{i\sqrt{3}}{2}$, confirming the solutions.

**Final Answer**: The solutions are $\boxed{-\dfrac{3}{2} \pm \dfrac{\sqrt{3}}{2}i}$.

## Example 2: Letter Counting in Misspelled Words (Part 1)

**Query:** how many r's in "strawberrry"?

*Model's reasoning process:*

> **Think**
>
> Okay, let's see. The question is asking how many times the letter "r" appears in the word "strawberrry". Hmm, first I need to make sure I spell the word correctly. The user wrote "strawberrry". Let me check that. Normally, the plural of "strawberry" is "strawberries", but maybe they made a typo or a spelling mistake. Anyway, the word given here is "strawberrry".
> Let me count the letters one by one.
> Breaking it down: s-t-r-a-w-b-e-r-r-r-y. Let's write each letter separately:
> S, T, R, A, W, B, E, R, R, R, Y.
> Wait, so starting from the beginning: S (1), T (2), R (3), A (4), W (5), B (6), E (7), R (8), R (9), R (10), Y (11). So total letters are 11. Now, how many times does R appearhere?
> Looking at the letters: positions 3, 8, 9, 10. That's four R's. Let me count again to be sure.
> S-T-R-A-W-B-E-R-R-R-Y. So after E comes R, then another R, another R, and then Y. So that's three R's in a row? Wait, no. Let's see: after E is the first R, then the second R, third R, and then Y. Wait, no, the word is "strawberrry". So after E, there are three R's in a row, right? Because it's written as "berrry". So breaking down:
> s-t-r-a-w-b-e-r-r-r-y. So after E, there are three R's before Y. So that's three R's. Wait, but when I listed them earlier, I counted four R's. Let me check again.
> Wait, maybe I miscounted. Let's write out each letter with their positions:
> 1. S 2. T 3. R 4. A 5. W 6. B 7. E 8. R 9. R 10. R 11. Y
> So positions 3,8,9,10 are R's. That's four R's. But wait, the word is spelled as "strawberrry".

*To be continued...*

## Example 2: Letter Counting in Misspelled Words (Part 2)

*Model's reasoning process (continued):*

### Think

So after "straw" and "b", it's "errry". So "e", then three R's, then "y". So that's three R's. But according to the breakdown above, there are four R's. Wait, maybe the user added an extra R by mistake? Let me check the exact spelling again. The user wrote "strawberrry". So "berrr". So "b", then "e", then three R's, then "y". So that's three R's. But in my positional list, I have four R's. Wait, maybe I made an error in counting.

Wait, let's write the word step by step:

s (1) t (2) r (3) a (4) w (5) b (6) e (7) r (8) r (9) r (10) y (11)

So letters 3,8,9,10 are R's. That's four R's. But the user wrote "strawberrry", which would be "berrr" at the end. So "b", "e", "rrr", "y". So three R's. But according to the breakdown, it's four. Wait, maybe the user intended "strawberry" but misspelled it as "strawberrry". Let me confirm. The correct plural of "strawberry" is "strawberries", which has two R's. But the user wrote "strawberrry", which has three R's in a row. So perhaps the answer is three? Or four?

Wait, maybe I'm overcomplicating. Let's just take the word as given: "strawberrry". Let's count each letter:

s, t, r, a, w, b, e, r, r, r, y. So that's 11 letters. The R's are at positions 3,8,9,10. So four R's. Therefore, the answer should be 4. But wait, maybe the userintended "strawberry" with two R's, but wrote "strawberrry" by accident. But the question is about the given word, so we have to go with that. So in "strawberrry", there are four R's. Wait, but let me check again. Let's write the word out:

s t r a w b e r r r y

Yes, after the E, there are three R's. So that's three R's. Wait, but in the list above, positions 3,8,9,10. Wait, position 3 is the third letter, which is R. Then after E (position 7), positions 8,9,10 are R's, and Yis 11. So that's three R's after the E. But position 3 is an R as well. So total R's are four. Wait, but in the word "strawberrry", the first R is in "straw", then after "straw", there's "b", "e", then three R's. So total R's would be 1 (from "straw") plus 3 (from "berrry") equals 4. So the answer is 4. But maybe I'm miscounting. Let me write it again:

s (1) t (2) r (3) a (4) w (5) b (6) e (7) r (8) r (9) r (10) y (11)

Yes, that's four R's. So the answer is4. But maybe the user intended a different count. Alternatively, maybe the word is spelled differently. Wait, "strawberry" is spelled s-t-r-a-w-b-e-r-r-y, which has two R's. But the user wrote "strawberrry", which adds an extra R, making it three R's in a row. Wait, no: "strawberrry" would be s-t-r-a-w-b-e-r-r-r-y, so after "ber" there are three R's. So total R's: in "straw" there's one R, then "berrry" has three R's. So total 1 + 3 = 4. Therefore, four R's. So the answer is4. But I need to make sure. Let me check the exact spelling again. The user wrote "strawberrry". Let's break it down:

straw + berry, but with rrrr instead of rr. So "straw" has one R, "berry" normally has two R's, but here it's written as "rrr", which would be three R's? Wait, no. Wait, "berry" is b-e-r-r-y. So two R's. If it's written as "berrry", that's replacing the double R with triple R? Wait, no. Let me think. The correct plural is "strawberries", which is s-t-r-a-w-s-b-e-r-r-i-e-s. But perhaps the user made a typo. However, the question is about the given word "strawberrry". So regardless of the correct spelling, we have to count the R's in that exact spelling.

So "strawberrry" is s-t-r-a-w-b-e-r-r-r-y. Let's count the R's:

1. The R in "straw" (third letter) 2. The R in "ber" (eighth letter) 3. The next R (ninth) 4. The next R (tenth)

So four R's. Therefore, the answer is 4. But maybe I'm missing something. Let me check oncemore. Let's write each letter with its case, but the problem didn't specify case sensitivity, so assuming all lowercase aswritten. So the letters are s, t, r, a, w, b, e, r, r, r, y. So R's at positions 3,8,9,10. Four of them. So the answer should be 4. But wait, sometimes people might confuse the number of R's in a double letter. For example, "berrry" might be considered as three R's, but in this case, the word is "strawberrry" with three R's after "ber". So yes, three R's there. Plus the one in "straw" makes four. So the answer is 4. I think that's correct.

**Model's Response:**

The word "strawberrry" is spelled as s-t-r-a-w-b-e-r-r-r-y. Breaking it down:

1. **s** 2. **t** 3. **r** 4. **a** 5. **w** 6. **b** 7. **e** 8. **r** 9. **r** 10. **r** 11. **y**

The letter **r** appears at positions 3, 8, 9, and 10, totaling **4** occurrences.

4

# H    Additional Ablation Study

Table 9: Ablation study results on Phonebook 32K (PB-32k), language modeling and common-sense reasoning tasks in zero-shot for 1B-parameter models with a sliding window size of 128. The average accuracy does not include PB-32K. NB means applying normalization before output gating in the last Mamba-2 layer of self-decoder, GMU means replacing nGMU with GMU and using the memory after normalization but before output gating.

| Model | Speed mtps ↑ | Wiki. ppl ↓ | PB-32K acc ↑ | LMB. acc ↑ | ARC-c acc_n ↑ | ARC-e acc ↑ | Hella. acc_n ↑ | PIQA acc ↑ | Wino. acc ↑ | Avg. acc ↑ |
|---|---|---|---|---|---|---|---|---|---|---|
| SambaY-2 | 1.43 | 17.17 | 40.63 | 48.96 | 28.84 | 59.18 | 48.01 | 70.18 | 50.83 | 51.00 |
| w/ NB + GMU | 1.40 | 17.76 | 21.88 | 49.49 | 29.69 | 59.68 | 48.71 | 71.22 | 52.17 | 51.83 |
| MambaY-2 | 1.38 | 18.63 | 50.78 | 49.58 | 28.24 | 58.75 | 48.29 | 70.13 | 51.07 | 51.01 |
| w/ NB + GMU | 1.35 | 16.99 | 17.19 | 49.76 | 27.39 | 58.46 | 48.43 | 70.24 | 50.28 | 50.76 |
| S-GDNY | 1.34 | 16.78 | 83.59 | 50.94 | 29.61 | 58.96 | 48.93 | 71.55 | 51.85 | 51.97 |
| w/ GDN + GMU | 1.33 | 16.84 | 27.34 | 51.08 | 28.16 | 57.49 | 48.25 | 69.15 | 53.04 | 51.20 |
| GDNY | 1.22 | 16.92 | 89.84 | 50.38 | 28.84 | 60.61 | 48.01 | 71.27 | 51.38 | 51.75 |
| w/ GDN + GMU | 1.24 | 16.87 | 54.69 | 50.30 | 27.73 | 60.48 | 47.91 | 70.62 | 52.17 | 51.53 |

**How does normalization placement and nGMU affect the model performances?**    Table 9 reveals a consistent pattern: retaining the nGMU and applying RMSNorm *after* the output gating is critical for long-context retrieval performance. In contrast, shifting the normalization *before* the gate and replacing nGMU with the simpler GMU ("NB + GMU" rows) leaves short-context benchmarks largely unaffected but leads to severe performance degradation on PB-32K across all linear-attention variants. For example, PB-32K accuracy drops by 56.3 points for S-GDNY (from 83.6 to 27.3) and by 35.1 points for GDNY (from 89.8 to 54.7), despite minimal changes (≤3%) in Wiki perplexity, zero-shot commonsense scores, and throughput. These results underscore the importance of maintaining the associativity between gating and token mixing by (1) normalizing *after* the output gating and (2) using memory before normalization with nGMU for achieving effective long-range retrieval performance with linear attention layers in self-decoder.

# I    Related Work

**KV Cache Sharing.**    Efficient inference in transformer-based models has been significantly advanced through techniques that reduce memory consumption, particularly concerning key-value (KV) caching. Traditional approaches like Multi-Query Attention (MQA) [Sha19] and Grouped-Query Attention (GQA) [ALTdJ+23] have enabled multiple query heads to share a single key/value head within the same layer, effectively reducing the number of distinct key/value caches with minimal impact on accuracy. Apart from YOCO [SDZ+24], Cross-Layer Attention (CLA) [BMN+24] extends KV sharing across adjacent layers, achieving up to two times reduction in KV cache size while maintaining performance. In contrast, our work studies representation sharing across SSM/RNN layers, and proposes to directly share the output from the SSM kernel to avoid materializing recurrent states, thereby preserving the parallel training efficiency of linear recurrent models.

**Efficient Long Generation.**    Efficient long-sequence generation in transformer models has been a focus of recent research on LLM efficiency, primarily due to the substantial memory demands associated with key-value (KV) caching during inference with long CoTs [KKL20, WWZ+24, YCQ+21, DYL+24]. To address these challenges, several techniques have been proposed to optimize memory usage without compromising model performance. One notable approach is the Layer-Condensed KV Cache (LCKV) [WT24], which computes and caches KV pairs for only a subset of layers, significantly reducing memory consumption and improving inference throughput. Another advancement is InfiniGen [LLSS24], a dynamic KV cache management framework that selectively prefetches essential KV cache entries, thereby mitigating fetch overhead from host memory in offloading-based LLM serving systems. These methods collectively contribute to more efficient long-sequence generation by optimizing KV cache usage, and are orthogonal to our work, as we can also apply these techniques to improve the memory I/O efficiency of our full attention layer.

**Hybrid Neural Architectures.**    Recent hybrid models have explored combining different types of token mixing operators—including Sliding Window Attention (SWA) [BPC20], full self-attention

[VSP+17] and SSMs/RNNs—either in an inter-layer [FDS+22, DSF+24, LLB+24, RLL+25, Min25] or an intra-layer manner [MZK+22, RLW+23, PFF+23, MYX+24, MFG24, DFD+25]. As a typical design of intra-layer hybridization, the efficiency of hybrid-head architecture [WRHS22, ZLJ+22, MFG24, DFD+25] is bottlenecked by the slowest token-mixing head, resulting in theoretically lower GPU utilization than inter-layer hybridization. Samba [RLL+25], an inter-layer hybrid model that interleaves Mamba with SWA, achieves improved extrapolation perplexity on extremely long sequences while maintaining linear complexity. However, its zero-shot retrievable context length remains limited to its sliding window size. The decoder-decoder architecture, YOCO [SDZ+24], proposes to use linear complexity modules (either SSMs or SWA) in the first half of the layers with a single full attention in the middle, and reuse the kv cache of the middle full attention layers for the second half of the attention layers. It shows comparable performance on retrievable context length as the full attention models while providing much more efficient linear pre-filling complexity. This design also offers a unique advantage that allows skipping inference computation in half of the total layers at the prefill stage, yielding substantial efficiency gains—even for short sequences, where MLPs dominate the computational cost. Our proposed GMU module opens up new opportunities for the pure RNN-based models to be YOCO-compatible, potentially mitigating the significant overhead that linear RNNs typically incur on short sequences.

**Neural Scaling Laws.**   Understanding how model performance scales with size and data is crucial for efficient and effective large-scale training. Empirical studies have shown that Transformer models exhibit predictable scaling behaviors, where performance improves with increased model parameters and training data [HNA+17, KMH+20, BDK+24, ANZ22, HBM+22]. Numerous works have also investigated scaling laws for hyper-parameters, based on either empirical studies [BBC+25, WLX+24] or theoretical analyses [MLPA22, YHB+22, YYZH23, WA24]. In this work, we focus on theoretical hyper-parameter scaling laws since they are not over-tuned for the Transformer architectures, and fairer comparisons can be made for the emerging neural architectures. We also conduct extensive scaling experiments with large-scale compute to verify the empirical effectiveness of these theoretical scaling laws. In doing so, we find an improved version of the original $\mu$P [YHB+22] that accounts for scaling of depth, width, and training stability, and demonstrate that it provides better scaling behavior in both data and compute scaling scenarios. More importantly, we introduce a principled approach for comparing the scaling behaviors of different neural architectures by solving iso-parametric equations, providing a solid foundation for evaluating the scaling potential of future architectures.

## J   Limitation

We validate our model's reasoning capability using distillation-based Supervised Fine-Tuning (SFT), but Reinforcement Learning (RL) remains under-explored in the context of hybrid architectures. Due to resource constraints, we do not perform an exhaustive hyperparameter search for each architecture. Instead, we adopt a generic optimization setup based on Transformer++ for learning rate, initializer range, weight decay, warm-up schedule, batch size, AdamW betas and epsilon, and other parameters. It is likely that aggressive tuning of these optimization settings could yield improved results. We leave a more comprehensive study of the interplay between optimization setups and architecture designs for future work. Lastly, our architecture still includes a full-attention layer, which leads to linear per-token computation complexity during decoding. This underscores a future research direction on designing models for extremely long sequence generation that can maintain constant decoding complexity while effectively leveraging long-context memory.

