# OpenReview forum: "Decoder-Hybrid-Decoder Architecture for Efficient Reasoning with Long Generation"
_NeurIPS.cc/2025/Conference — NeurIPS 2025 poster_

### Official Review · Reviewer_JjzD · 2025-06-20

**Clarity:** 3
**Significance:** 2
**Originality:** 2
**Rating:** 4
**Confidence:** 4

**Summary:**

This paper addresses the efficiency limitations of Transformer models when handling long sequences by proposing a hybrid architecture that integrates a state-space model (SSM) into a decoder-decoder framework. Specifically, the authors enhance the YOCO architecture by replacing its self-decoder with Samba, a hybrid SSM, and introduce a gated memory unit (GMU) to improve cross-decoding. The resulting model, SambaY, is evaluated across several tasks:

- Scaling on NLP tasks: SambaY demonstrates superior FLOPs scaling compared to Transformer baselines. However, the GMU offers limited additional benefits.
- Long-context retrieval: SambaY outperforms Transformer baselines, with the GMU and differential attention (DA) contributing positively. The GMU, however, only gives minor benefits on other downstream tasks (Wiki, ARC-c, etc.).
- Large-scale pretraining: A scaled version of SambaY (Phi4-mini-Flash) trained on proprietary data outperforms Phi4-mini on downstream tasks.
- Reasoning tasks: Phi4-mini-Flash matches Phi4-mini-Reasoning in performance while improving throughput.

**Questions:**

I would appreciate if the additional experiments (mentioned in Strengths and Weaknesses) could be included in the rebuttal. Moreover, I have the following comments/questions:
- Line 164 states “This aligns with our expectation, as there is no theoretical expressiveness gap between Transformers and our hybrid models when the number of parameters is held constant.” This statement is very strong and requires more investigations (e.g., [Merill et al, ICML 2024]).
- Line 288 states “Our extensive scaling experiments demonstrated that SambaY
exhibits a lower irreducible loss compared to strong baselines, indicating superior scaling properties with increasing computational resources.” This statement is clearly conditioned on the modelling/fitting of the loss function. Hence, it should also be stated that way.

**Ethical Concerns:**

["NO or VERY MINOR ethics concerns only"]

**Limitations:**

While the second point in the NeurIPS Paper Checklist states that the limitation is included in the conclusion section, they are instead (a bit hidden) in the appendix. It would be better to have limitations in the main paper.

**Quality:**

3

**Strengths And Weaknesses:**

Strengths:
- The paper presents a comprehensive experimental evaluation across multiple tasks and scales.
- The integration of SSMs into a decoder-decoder architecture is well-motivated and practically relevant.

Weaknesses:
- **Missing baselines** As this work mainly builds on top of YOCO, it should be considered as a baseline in the experiments (compute and data scaling experiments, retrieval, and downstream tasks). Otherwise, we cannot see the effect when going from SWA to Mamba in the self-decoding. Moreover, Table 4 should also include the Phi-mini-Flash (without reasoning SFT).
- **Mixed results of the gating mechanism (GMU)** The experiments fail to clearly demonstrate the benefits of GMU. In many benchmarks, just merging Samba with YOCO (Samba+YOCO) is on par with SambaY. Indeed, the performance of SambaY (which uses GMU) had to be pushed with differential attention (DA). In that regard, testing Samba+YOCO+DA may even give the best results.

---

> ### Author Response · Authors · 2025-08-04
>
> We would like to thank the reviewer for the positive review of our work and the constructive feedback on our manuscript. In the following, we address the remaining concerns to hopefully motivate a clear acceptance score.
>
> **More baselines:** Thanks for the suggestions. Due to the limited time and computation resources, we are unable to draw the expensive scaling curves (which needs more than 12K H100 hours for the FLOPs scaling experiment) during the rebuttal time. However, we do conduct the long context retrieval experiments as follows.
>
> We pre-train 1.0B parameter SWA+YOCO models with µP++ and d = 16 on ProLong-64k dataset with 40B tokens. The SWA+YOCO architecture uses all SWA layers for self-decoder and all cross-attention layers for cross-decoder.  We align the parameters through solving the iso-parameteric equation to obtain the aspect ratio $\alpha = 130$. A 640K RoPE base is also adopted. The following table shows the PB-32K evaluation results of six models trained with various sliding window sizes. We can see that SWA+YOCO achieves the best result with a SWA size of 128, while its performance is consistently worse than Samba+YOCO or SambaY.
>
> | SWA Size | 64  | 128 | 256   | 512  |  1024 | 2048 |
> |--------|------|-----|--------|-----|--------|----|
> | SWA+YOCO |  13.28 ±  3.40 | 37.50  ± 4.42  |  30.47 ± 7.45    |  13.28  ±  2.59   |  16.41 ±   4.62  |   8.59 ± 4.62   |
>
>
> We also measure the NIAH tasks performance in the RULER benchmarks as follows. We can see that SWA+YOCO with its best 128 SWA size provides the worst average accuracy across all models.
>
> | Model          | SWA  | MK-1 | MK-2 | MK-3 | MQ    | MV    | S-1   | S-2   | S-3   | Avg. |
> |-----|------|------|------|------|-------|-------|-------|-------|-------|-------|
> | Transformer++  | -    | 36.4 | 3.8  | 0.0  | _27.9_ | _24.1_ | 94.8  | 66.0  | 31.0  | 35.5  |
> | TransformerLS  | 256  | 42.8 | 6.0  | 0.0  | **29.8** | **27.5** | 91.8  | 49.6  | 23.4  | 33.9  |
> | Samba+YOCO     | 1024 | 49.0 | **28.0** | **2.6** | 12.8  | 18.3  | **100.0** | 63.2  | 23.6  | 37.2  |
> | SambaY       | 256  | _54.6_ | _27.8_ | _0.4_ | 12.7  | 19.4  | 83.2  | _81.2_ | _63.8_ | _42.9_ |
> | SambaY+DA  | 512  | **64.6** | 27.6 | 0.2  | 12.8  | 19.9  | _99.8_ | **86.4** | **69.6** | **47.6** |
> | SWA+YOCO| 128  |24.2| 6.8| 0.2 | 10.2 | 14.7 | 81.2|32.6 |48.4| 27.3 |
>
> We further measure the downstream short-context performance on language modeling and common-sense reasoning tasks in zero-shot, and we can see that SWA+YOCO is comparable to TransformerLS but has worse performance than Samba+YOCO or SambaY.
>
>
> | Model         | SWA  | Speed (mtps ↑) | Wiki. (ppl ↓) | LMB. (acc ↑) | ARC-c (acc_n ↑) | ARC-e (acc ↑) | Hella. (acc_n ↑) | PIQA (acc ↑) | Wino. (acc ↑) | Avg. (acc ↑) |
> |------|------|-----|--------|---------------|----------|----------------|----------|---------|----------|-------|
> | Transformer++ | -    | 0.89           | 19.75         | 45.45         | 27.56           | 54.17         | 43.86            | 68.77         | 50.28         | 48.35          |
> | TransformerLS | 256  | **1.46**       | 18.49         | 48.77         | _28.84_         | 57.11         | 45.85            | 69.21         | _53.67_       | 50.57          |
> | Samba+YOCO    | 1024 | 0.99           | _16.73_       | **50.53**     | 28.50           | _60.02_       | 48.85            | **71.55**     | 52.57         | 52.00          |
> | SambaY        | 256  | 1.11      | 17.83         | _50.40_       | **29.44**       | 57.87         | _49.08_          | 71.00         | **55.25**     | **52.17**      |
> | SambaY+DA     | 512  | 0.91           | **16.59**     | 49.68         | 28.33           | **60.65**     | **49.53**        | _71.38_       | 53.43         | **52.17**      |
> | SWA+YOCO| 128 | _1.24_ | 18.01  | 49.80 | 28.58   | 57.79   | 46.48    | 70.46 | 51.85   | 50.69  |
>
>
> **GMU Performance:** Thank you for the detailed analysis. While performance on some short-context benchmarks may appear comparable, this perspective overlooks the two crucial advantages that the Gated Memory Unit (GMU) provides to our SambaY architecture: (1) The primary motivation for the GMU is to reduce the severe memory I/O bottleneck present in standard cross-attention during decoding. This is where SambaY excels. As shown clearly in Figure 4, SambaY delivers substantially higher throughput than Samba+YOCO. (2) On tasks specifically designed to test long-context retrieval, SambaY demonstrates a clear performance advantage over Samba+YOCO. In the challenging Needle-in-a-Haystack (NIAH) benchmark from RULER (Table 1), SambaY achieves an average accuracy of 42.9, significantly outperforming Samba+YOCO's score of 37.2.
>
> **Limitations:** Thanks for the suggestion. Due to the page limitations, we will add a reference at the beginning of section 3 to guide the reader for the limitation section in the appendix.
>
> *(to be continued)*

---

> > ### Author Response · Authors · 2025-08-04
> >
> > *(continued)*
> >
> > ---
> >
> > **Q1:** *Line 164 states “This aligns with our expectation, as there is no theoretical expressiveness gap between Transformers and our hybrid models when the number of parameters is held constant.” This statement is very strong and requires more investigations (e.g., [Merill et al, ICML 2024]).*
> >
> > **A1:**  Thanks for the thoughtful comments. The Mamba-1/2 used in this work do indeed fall into the same complexity class of TC^0 as Transformers, so that’s the reason why we claim our hybrid models have the same expressiveness power as Transformers. We will modify the statement as follows: “This aligns with our expectation, as there is no theoretical expressiveness gap between Transformers and our hybrid models when the number of parameters is held constant. This is because we use Mamba-1 as our SSMs, which falls into the same complexity class of TC^0 as Transformers. [Merill et al, ICML 2024]”
> >
> >
> > **Q2:** *Line 288 states “Our extensive scaling experiments demonstrated that SambaY exhibits a lower irreducible loss compared to strong baselines, indicating superior scaling properties with increasing computational resources.” This statement is clearly conditioned on the modelling/fitting of the loss function. Hence, it should also be stated that way.*
> >
> > **A2:**  Thanks for the suggestion. We will modify the statement as follows: “Our extensive scaling experiments demonstrated that SambaY exhibits a lower irreducible loss compared to strong baselines when fitted with power laws against FLOPs, indicating superior scaling properties with increasing computational resources.”

---

> ### Author Response · Authors · 2025-08-05
>
> Dear reviewer JjzD,
>
> As we approach the end of the discussion period, we would greatly appreciate your input on the paper. We hope our responses and additional results address your concerns and welcome any further questions or suggestions.
>
> Thank you for your time and effort.

---

### Official Review · Reviewer_RQ42 · 2025-07-01

**Clarity:** 3
**Significance:** 3
**Originality:** 3
**Rating:** 5
**Confidence:** 4

**Summary:**

First, this paper introduces the Gated Memory Unit (GMU), a neural network layer for accessing memories of other/previous layers.
Second, the GMU is then applied to a new Hybrid Decoder-Decoder architecture called SambaY, which uses Samba (a hybrid Mamba Attention architecture) for the self decoder and replaces half of the cross-attention layers with GMUs to access the inner representations of the SSM state of the Samb self decoder layer.
Finally, the paper introduces Phi4-mini-Flash, a 3.8B model trained on 5T tokens, which replaces the attention layers in SambaY with Differential Attention (DA).

The post-trained version Phi4-mini-Flash-Resoning achieves comparable performance to the Phi4-mini baseline on reasoning tasks while delivering higher decoding throughput.

**Questions:**

- How do you explain the drop in performance on PB-32k when moving from SambaY to SambaY-2?
It seems that without SWA Mamba2 is better, while with SWA Mamba1 is better.
This seems to contradict the explanation in L. 263.
( Could it be that the reason is a less effective memory access to matrix states with GMU? (see next question) )
- How do you apply the GMU to memory states which are matrices instead of vectors (i.e. Mamba2 has a matrix memory state)? Do you flatten the matrices?

- L.279: „Gating attention/MLP representations performs worse … because they lack recency bias ..“: Why is the recency bias in the memory state necessary/helpful for GMU?

- Is the fitting of the powerlaw in Figure 2 performed in logspace?
- How does the SambaY or SambaY-DA architecture scale with number of parameters beyond 3.8B paramaters? Since the authors allocated a significant amount of compute to train the 3.8B model on 5T tokens, why were larger models not explored? Is there anything that prevents scaling to larger models?

**Ethical Concerns:**

["NO or VERY MINOR ethics concerns only"]

**Final Justification:**

The authors addressed all my questions. Therefore, I will keep my score.

**Limitations:**

yes

**Paper Formatting Concerns:**

No concerns.

**Quality:**

3

**Strengths And Weaknesses:**

Strengths:
- Extensive and large-scale experiments on scaling behavior and good scaling performance
- The scaling setup is explained in detail and the relations between width, depth, and hyperparameters are given as formulas.
- Detailed ablation study on long context performance
- Superior performance compared to Transformer++ baseline on Long-Context Benchmarks

Weaknesses:
- In the ablation study in Table 5, the Differential Attention used in the final Phi4-mini-Flash model is missing
- The authors observe significantly better scaling behavior with their µP++ parameterization over SP. Since µP++ is a (new) combination from µP and Depth-µP it would be helpful to clarify the differences to SP in more detail, i.e. how does SP look like in this paper and what was modified going from SP to µP++.

---

> ### Author Response · Authors · 2025-08-04
>
> We would like to thank the reviewer for the very positive review of our work and the interesting questions on our manuscript. We address these questions in the following paragraphs.
>
> **Ablation Studies with Differential Attention:** Since our primary focus is on ablating the architectural design of the self-decoder and cross-decoder components in the SambaY architecture, we did not include Differential Attention (DA) in the main ablation results. DA is a simple, orthogonal extension to SambaY and does not directly impact the architectural comparisons we aim to study. However, for completeness, we do include results with DA in Figure 3 and Table 2 for ablating the design of Phi4-mini-Flash.
>
> **Differences between SP and µP++:** Thanks for the suggestion! We summarize the key differences between μP, μP++, and Standard Parameterization (SP) in the following table.  *LR mult.* denotes the per-parameter multiplier applied on top of the global learning rate (η), *Res. mult.* is the multiplier applied to the output of residual branches, and *WD* denotes the weight decay. For μP++, η ∝ 1/√d and zero weight decay is also applied to other scalar or vector-like parameters such as RMSNorm weights. In this work, σ = 10⁻⁴ for untied embedding and σ = 0.02 for tied embedding; in both cases, τ = 0.02 and β = 1. “fan_in” refers to the input dimension of weight matrices.
>
> | Parameter   | Scheme   | LR mult.     | Initialization      | Res. mult.   | Weight mult.     | WD      |
> |------|----------|----------|------------|----------|----------|-----------|
> | **Embedding**     | SP       | ∝ 1              | 𝒩(0, σ²)  | —                   | ∝ 1               | ∝ 1              |
> |        | μP       | ∝ 1     | 𝒩(0, σ²)           | —                   | ∝ 1               | ∝ 1              |
> |     | μP++     | ∝ 1     | 𝒩(0, σ²)       | —                   | ∝ 1               | 0                |
> | **Unembedding**   | SP       | ∝ 1     | 0 or tied       | —                   | ∝ 1               | ∝ 1              |
> |       | μP       | ∝ 1    | 0 or tied         | —     | ∝ 1/w             | ∝ 1              |
> |        | μP++     | ∝ 1     | 0 or tied    | —       | ∝ 1/w             | 0                |
> | **Hidden Weights**| SP       | ∝ 1     | 𝒩(0, τ²)        | 1        | ∝ 1     | ∝ 1              |
> |      | μP       | ∝ 1/w      | 𝒰(−β/√fan_in, β/√fan_in)        | 1      | ∝ 1      | ∝ 1              |
> |      | μP++     | ∝ 1/w    | 𝒰(−β/√fan_in, β/√fan_in)       | 1/√2d     | ∝ 1               | ∝ 1              |
>
>
>
>
> ---
>
> **Q1:** *How do you explain the drop in performance on PB-32k when moving from SambaY to SambaY-2? It seems that without SWA Mamba2 is better, while with SWA Mamba1 is better. This seems to contradict the explanation in L. 263. ( Could it be that the reason is a less effective memory access to matrix states with GMU? (see next question) )*
>
> **A1:** Thank you for the insightful question! We believe the performance difference is best understood from a capacity trade-off perspective. Mamba-1 provides strong recency bias through fine-grained memory decay, enabling effective local representation and positional encoding. Mamba-2, on the other hand, uses a larger matrix-valued recurrent state, which enhances memorization capacity but weakens recency bias. When combined with SWA, which already offers strong local memorization and retrieval, the positional representation power of Mamba-1 becomes more valuable. In this setting, replacing Mamba-1 with Mamba-2 (as in SambaY-2) leads to redundant memorization and a net loss in positional expressiveness, hurting performance on tasks like PB-32k that benefit from precise local memorization. Conversely, in the absence of SWA, the increased memorization capacity of Mamba-2 in self-decoder becomes advantageous, enhancing the long-context retrieval of complex information (such as phone numbers) for the cross-decoder, where local memory support from SWA is missing. We will include this discussion in the next version of our paper.
>
> *(to be continued)*

---

> > ### Author Response · Authors · 2025-08-04
> >
> > *(continued)*
> >
> > **Q2:** *How do you apply the GMU to memory states which are matrices instead of vectors (i.e. Mamba2 has a matrix memory state)? Do you flatten the matrices?*
> >
> > **A2:**  While Mamba-2 does have an internal matrix-valued memory state, our Gated Memory Unit (GMU) is not applied to this internal state, but operates on the output hidden state of the SSM kernel. In a Mamba-2 block, the internal matrix state is used to compute an output vector, which is then gated and linearly projected to be added back to the main branch. It is this output vector that we designate as the "memory" ($\mathbf{m}$ in our formulation) for the GMU.
> > This design choice ensures a consistent, vector-to-vector interface for the GMU across all our model variants, regardless of the memory source's internal structure, meanwhile allowing GMU to reweight the past token-mixing matrix based on the currently layer input directly (Line 75-77) due to the associativity of the matrix multiplication. Therefore, there is no need to flatten any matrices, as the GMU always receives a standard hidden state vector as its memory input.
> >
> >
> > **Q3:** *L.279: „Gating attention/MLP representations performs worse … because they lack recency bias ..“: Why is the recency bias in the memory state necessary/helpful for GMU?*
> >
> > **A3:**  Thank you for the thoughtful question! The recency bias in the memory state is important for GMU because it enables more effective retrieval of recent queries from the self-decoder (such as the queried name in the phonebook task). By emphasizing recent information, the SSM component helps GMU preserve relevant query context, allowing the subsequent cross-attention layer to retrieve the correct memory entry based on the query representation produced by the previous GMU layer. This recency-aware retrieval serves as a valuable complement to the cross-attention mechanism. We will include this clarification in the next version of our paper.
> >
> >
> >
> >
> > **Q4:** *Is the fitting of the powerlaw in Figure 2 performed in logspace?*
> >
> > **A4:**  No, as said in Line 143, it is in the original space with non-linear least squares fit, following the setting of Chinchilla Law. In this way, we can have a good estimate of the irreducible loss C.
> >
> > **Q5:** *How does the SambaY or SambaY-DA architecture scale with number of parameters beyond 3.8B paramaters? Since the authors allocated a significant amount of compute to train the 3.8B model on 5T tokens, why were larger models not explored? Is there anything that prevents scaling to larger models?*
> >
> > **A5:**  We are a small research team funded with only 1K A100 GPUs with a limited amount of time to study neural architectures only for small language models with less than 4B parameters.  We didn’t see anything preventing scaling it further based on the current empirical results and are very interested in scaling it to larger dense or MoE models if more computation resources and people can be allocated to our team.

---

### Official Review · Reviewer_vaxF · 2025-07-02

**Clarity:** 4
**Significance:** 4
**Originality:** 3
**Rating:** 5
**Confidence:** 4

**Summary:**

The paper introduces SSMs to the space of decoder-decoder architectures such as YOCO. It introduces a new Gated Memory Unit (GMU) to efficient memory sharing across layers. This, when introduced into the cross-decoder of the base YOCO architecture (modified with Samba SSMs), creates a new decoder-hybrid-decoder architecture, referred to as Samba. Since the model shares KV cache across the decoder and cross-decoder, it helps ensure that prefill is linear in complexity (since only last full attention computes the KV cache for rest of cross-decoder). The authors then follow this up with rigorous training experiments, including µP++ - which combines both regular µP and depth-µP, batch size scaling and data scaling to show scaling behaviors (through power law curve fits) of the models in comparison to transformers and its variants transformer++ - both with SP (standard trianing) and µP++. They follow this up with long context extension (& evaluation) to show that the model scales well for inference (better than transformers, just slighlty worse relative to pure Samba architectures). They finally follow this up by scaling to a new model, Phi4-mini-flash, including large scale pre-training + SFT for reasoning.

**Questions:**

N/A

**Ethical Concerns:**

["NO or VERY MINOR ethics concerns only"]

**Paper Formatting Concerns:**

The references are in a different format relative to the suggested way in the style guide. Requesting authors to re-check this and ensure consistent formatting.

**Quality:**

4

**Strengths And Weaknesses:**

**Strengths**

1. The architecture introduces GMU, and shows how the modifications help with modeling
2. The paper throughly explores all axes (scaling model size, dataset size, pre-training haprams, long-context retrieval, downstream evaluation)
3. The different architecture ablations are useful to understand the design choices of the model.
4. The paper also shows inference scaling (through random weights in vLLM) to showcase how the linear-time complexity behavior they talk about.

**Weaknesses**

1. For some of the architecture ablations, it would have been good to see better downstream tasks for evaluation. Most of the tasks selected are very generic language modeling tasks with relatively high variance in scores - making the presented results a little tricky to parse.

---

> ### Author Response · Authors · 2025-08-04
>
> We would like to thank the reviewer for the very positive review of our work. We address the remaining concerns in the following paragraphs.
>
> **Downstream Evaluation:** Thanks for your suggestions! Our primary objective in the ablation studies was to test the efficiency of different memory-gating strategies for long-context retrieval. To this end, we deliberately chose the Phonebook (PB-32K) benchmark. This is not a generic language modeling task but is a synthetic, highly-controlled test designed to provide a precise signal on a model's ability to perform multi-key retrieval. As shown in Table 5, the results on this task are not noisy but they reveal a clear and significant performance hierarchy (SambaY > SambaY-MLP > SambaY-A > SambaY-AA), which directly validates our core design choices.
>
> **Paper Formatting**: Thanks for pointing it out! We have double checked the style guide, which does not exclude the usage of the current formatting since we maintain the internal consistency.

---

### Official Review · Reviewer_j2N9 · 2025-07-03

**Clarity:** 1
**Significance:** 3
**Originality:** 2
**Rating:** 3
**Confidence:** 4

**Summary:**

This work leverages representation sharing between SSM layers in hybrid LLM architectures by gating states (memories) from earlier layers. To this end, the Authors propose the Gated Memory Unit (GMU). The Authors conduct extensive parameter and data scaling studies to compare the resulting architectures with competitive baselines. In a proof of concept based on the Phi4-mini family, they show performance gains in standard benchmarks and throughput increase in reasoning settings.

**Questions:**

- If YOCO reuses keys and values across layers, shouldn’t this also benefit decoding efficiency, as fewer values must be retrieved from GPU memory? (line 29)
- What is the advantage of removing RoPE in practice? This does not affect efficiency, and you do not provide empirical evidence that it mitigates some of RoPE’s possible challenges (e.g., length generalisation).
- Are you trying to claim that this architecture is superior for long context in general or just with respect to specific baselines? If the former, you would need to include state-of-the-art efficient transformer baselines, e.g., Quest sparse attention.
- How are the learning rate and batch size chosen in line 114?
- It might be just because of the image granularity, but how is C smaller for SambaY than Samba+YOCO in Figure 2b when the layer seems to display a consistent delta between the two lines?

**Ethical Concerns:**

["NO or VERY MINOR ethics concerns only"]

**Limitations:**

A more thorough discussion on how the usage of private data impacts reproducibility is needed.

**Quality:**

3

**Strengths And Weaknesses:**

**Strengths**
- The experiments are quite extensive and include parameter and data scaling studies for the considered architectures, with carefully scaled hyperparameters.
- The proofs of concept experiments on proprietary data are quite convincing. The gains of Phi4-mini-Flash, which incorporates the proposed GMU as well as pre-existing DA and other improvements, over Phi4-mini are substantial. Throughput gains over Phi4-mini-Reasoning in reasoning tasks are also noteworthy.

**Weaknesses**
- I am partly concerned about the novelty of the contributions and their coherence as part of a singular narrative. In particular:
    - GMUs, the µP++ hyper-parameter transfer scheme, and the Differential Transformer modification are given equal consideration (e.g., emphasis in the introduction); however, only the first one is novel to this paper.
    - To my understanding, the main contribution is extending YOCO’s idea (memory sharing across layers) from attention layers to SSM layers in a hybrid model. While this is a meaningful extension, it does not constitute a fully original idea.
- The main paper is not entirely self-contained: concepts such as YOCO, Cross-Decoder, SWA layers, etc., which the present approach builds on, should be described as part of the method or background and not taken for granted.
- The range of downstream tasks seems somewhat limited for long context, which is one of the main foci of the paper: why are only a subset of RULER tasks included (e.g., NIAH but no variable state tracking)?

**Minor**
- Why is the architecture defined in section 3.1 called Transformer++?
- The symbols in lines 99 and 111 should be explained to understand their meaning.
- In the ablation study in section 4, you list the model variants without the specific question/ablation each one is meant to test.

**Typos**
- “Stat[e]” (line 1)

---

> ### Author Response · Authors · 2025-08-01
>
> We would like to thank the reviewer for the constructive feedback on our manuscript. In the following, we address the remaining concerns to hopefully encourage a positive evaluation.
>
> **Novelty and coherence of the contributions:** We think that our work actually presents a cohesive narrative centered on our main contribution: the Gated Memory Unit (GMU), a novel and versatile architectural component that enables our efficient SambaY architecture. The GMU is our primary contribution, designed as a simple yet effective mechanism for sharing representations across layers. While inspired by the previous works on KV cache sharing, our work introduces a completely new way to achieve memory sharing for State Space Models (SSMs). As we show in L271-283, the GMU is a generalizable concept, capable of gating memory from attention and MLP layers, making it a distinct contribution beyond the specific context of YOCO (L75-78).
> We do not claim novelty for the µP++ scheme or Differential Attention (DA). Their roles are distinct and supportive: µP++ is a methodological tool for ensuring a rigorous and fair comparison between architectures. This was necessary to scientifically validate our claim that SambaY exhibits superior scalability with a lower irreducible loss (L42-43, L154-156). DA is an extension that demonstrates the power and versatility of our SambaY framework. By integrating it, we create a state-of-the-art model that proves SambaY's practical value and efficiency gains on challenging benchmarks (L54-55, L234-237).
> Therefore, our contributions are unified: we introduce a novel architecture (SambaY) enabled by a new component (GMU), use rigorous methodology (µP++) to prove its effectiveness, and showcase its full potential with a powerful extension (DA).
>
> **More backgrounds:** Thanks for pointing this out. We will add the following backgrounds in Appendix for more self-contained presentations:
>
> Our work builds upon the YOCO (You Only Cache Once), an efficient decoder-decoder architecture. It consists of a self-decoder, which is the first half of the layers that produces KV caches in linear timem and a cross-decoder, which is the second half of the layers that re-uses the KV caches from the first half. For an input sequence of hidden states $H_{mem} \in \mathbb{R}^{N \times d}$ from a single final attention layer in the self-decoder, it pre-computes and caches a single Key-Value (KV) pair:
> $$
> K_{c} = H_{mem}W_K, \quad V_{c} = H_{mem}W_V
> $$
> where $K_c, V_c \in \mathbb{R}^{N \times d_{kv}}$ are the cached matrices and $W_K, W_V$ are weight matrices. Subsequently, every cross-attention layer $l$ in the cross-decoder reuses this single KV cache. Given the input hidden state $H_{cross}^{(l)}$ to that layer, the cross-attention output is calculated by generating a new Query, $Q_{cross}^{(l)}$, and attending to the shared cache:
> $$
> Q_{cross}^{(l)} = H_{cross}^{(l)}W_Q^{(l)}
> $$
> $$
> \text{Output}^{(l)} = \text{softmax}\left(\frac{Q_{cross}^{(l)} K_c^T}{\sqrt{d_{kv}}}\right) V_c
> $$
> This approach avoids doing full attention computation at the prefill stage, reducing the prefill time complexity to linear.
>
> **More long-context evaluation:** Our selection of NIAH tasks was intentionally focused to evaluate the core hypothesis of Section 3.2 that SambaY can do efficient long-context retrieval. We also add additional evaluation on the remaining RULER tasks as follows. As we can see, all the models show near-zero Variable tracking (VT) performance because of the small scales of training (1B parameters with 40B tokens). We can also see that SambaY still has a better average accuracy than baselines.
>
> | Model| MK-1 | MK-2 | MK-3 | MQ   | MV   | S-1  | S-2  | S-3  | VT   | CWE  | FWE  | QA-1 | QA-2 | Avg.  |
> |----|----|----|----|----|----|---|---|----|----|----|----|----|----|----|
> | Transformer++ | 36.4 | 3.8  | 0.0  | 27.9 | 24.1 | 94.8 | 66.0 | 31.0 | 0.0  | 0.1  | 21.3 | 7.5  | 14.0 | 25.1  |
> | Transformer-LS | 42.8 | 6.0  | 0.0  | 29.8 | 27.5 | 91.8 | 49.6 | 23.4 | 1.0  | 1.0  | 2.9  | 5.3  | 13.8 | 22.7  |
> | Samba+YOCO   | 49.0 | 28.0 | 2.6  | 12.8 | 18.3 | 100.0| 63.2 | 23.6 | 0.0  | 3.2  | 47.7 | 9.4  | 13.6 | 28.6  |
> | SambaY  | 54.6 | 27.8 | 0.4  | 12.7 | 19.4 | 83.2 | 81.2 | 63.8 | 0.0  | 5.7  | 8.5  | 11.9 | 13.6 | 29.4  |
> | SambaY+DA | 64.6 | 27.6 | 0.2  | 12.8 | 19.9 | 99.8 | 86.4 | 69.6 | 0.2  | 4.3  | 8.9  | 12.7 | 15.2 | 32.5  |
>
> **Minor & Typos:**
> 1. *Why ...Transformer++?*
>
> This is because it improves over the original Transformer with RoPE and GLU, following the notation in [1].
>
> 2. *The symbols in lines ... understand their meaning.*
>
> N_{attn, mamba, mlp, gmu} is the total number of parameters for attention/Mamba/MLP/GMU layers respectively.
>
> *(to be continued)*

---

> > ### Author Response · Authors · 2025-08-01
> >
> > *(continued)*
> >
> > 3. *......you list the model variants without the specific question/ablation……*
> >
> > Thanks for pointing it out. We list the model details as a reference for the research questions we want to investigate in L260 and L271. We will present the variants more clearly in the next version.
> >
> >
> > 4. *“Stat[e]” (line 1)*
> >
> > Nice catch! Fixed it.
> >
> > ---
> >
> > **Q1:** *If YOCO reuses … as fewer values must be retrieved from GPU memory? (line 29)*
> >
> > **A1:** Not in practice. The limitation we highlight (line 29) occurs during the auto-regressive decoding stage. For every single token the model generates, each cross-attention layer in YOCO's cross-decoder must attend to the entire KV cache. This means that for a long prompt context of length N, the full Key and Value matrices must be loaded from slow GPU HBM into fast on-chip SRAM at every step, since SRAM is small and cannot hold all the KV cache for cross-layer re-use. This creates a significant memory I/O bottleneck.
> >
> >
> > **Q2:** *...... advantage of removing RoPE in practice? … (e.g., length generalisation).*
> >
> > **A2:**  Yes, we do observe better length generalization on long context retrieval for NoPE based models! In the following table, we measure the model accuracy at 32K, 64K and 128k context length on the Phonebook benchmark for 1B parameter models trained with 32K sequence length. We can see SambaY and its variants can extrapolate its retrieval ability by 2x in zero-shot, while rope based models (Transformer++ and TransformerLS) have a substantial drop beyond 32K.
> >
> > | Model         | SWA Size | 32k Acc (%)     | 64k Acc (%)     | 128k Acc (%)    |
> > |-----|---|-----|---|----|
> > | Transformer++ | -   | 60.94 ± 10.00    | 0.00 ± 0.00      | 0.00 ± 0.00      |
> > | TransformerLS | 256      | 60.16 ± 7.12     | 17.19 ± 5.63     | 0.78 ± 1.35      |
> > | Samba+YOCO    | 1024     | 82.81 ± 6.44     | 67.97 ± 11.13    | 20.31 ± 8.41     |
> > | SambaY         | 256      | 92.19 ± 1.56     | 96.09 ± 2.59     | 0.00 ± 0.00      |
> > | SambaY+DA    | 512      | 96.09 ± 1.35     | 84.38 ± 3.12     | 5.47 ± 2.59      |
> >
> >
> >
> > **Q3:** *Are you trying to claim that this architecture is superior for long context in general……*
> >
> > **A3:**  No, we just want to claim that it is better than the baseline architectures listed in the paper. Our contribution is orthogonal to the sparse attention variants.
> >
> >
> > **Q4:** *How are the learning rate and batch size chosen in line 114?*
> >
> > **A4:**  Thanks for pointing it out! We follow the setting in TinyLlama, and will add a reference in the next version.
> >
> > **Q5:** *......the layer seems to display a consistent delta between the two lines?*
> >
> > **A5:** Due to the figure’s granularity, the narrowing of the gap is not easily visible. However, it is indeed shrinking gradually, from an initial gap of 0.003 to a final gap of 0.001.
> >
> > ---
> >
> > [1] Transformer Quality in Linear Time (ICML 2022)

---

> > > ### Author Response · Authors · 2025-08-05
> > >
> > > Dear reviewer j2N9
> > >
> > > As we approach the end of the discussion period, we would greatly appreciate your input on the paper. We hope our responses and additional results address your concerns and welcome any further questions or suggestions.
> > >
> > > Thank you for your time and effort.

---

### Note · Authors · 2025-08-12

We sincerely thank all reviewers for their detailed feedback and constructive suggestions, which have helped us significantly improve our manuscript.

Throughout the discussion, we have aimed to clarify the novelty and coherence of our contributions. Our primary contribution is the Gated Memory Unit (GMU), a novel and versatile mechanism for memory sharing that includes but is not limited to State Space Models (SSMs), which is distinct from prior KV caching techniques. Our SambaY architecture, enabled by the GMU, demonstrates a key advantage over baselines like YOCO by improving long-context performance and substantially reducing the memory I/O bottleneck during decoding, leading to significantly higher throughput.

We bolstered our claims with new experiments during the rebuttal period. We presented additional long-context evaluations on all RULER tasks and introduced a SWA+YOCO baseline. The results confirm that SambaY achieves superior performance, particularly on challenging long-context retrieval benchmarks like Phonebook and Needle-in-a-Haystack. Furthermore, our new results on the Phonebook benchmark demonstrated the superior length extrapolation capabilities of our NoPE-based models.

We have also committed to adding more background on prerequisite architectures like YOCO, clarifying the nuances of our µP++ parameterization, and refining specific statements to better reflect the experimental evidence.

We are grateful for the opportunity to engage in this discussion and believe our responses and the additional results have addressed the reviewers' concerns. We hope our work is seen as a valuable contribution to developing more efficient and scalable language models for long generation.

---

### Decision · Program_Chairs · 2025-09-17

**Decision:**

Accept (poster)

**Comment:**

This paper presents a novel decoder-hybrid-decoder architecture, SambaY, aimed at improving the efficiency of large language models for long-generation reasoning tasks. The reviewers were broadly in agreement on the paper's primary strengths: a comprehensive and rigorous set of experiments, including detailed scaling studies, and impressive empirical results that demonstrate significant gains in both performance and inference throughput over baselines. However, a few weaknesses were also raised. The most significant concerns relate to the paper's core contribution, the Gated Memory Unit (GMU), as the experiments may not clearly isolate its benefits over simpler architectural variants. Furthermore, the lack of a direct comparison to the foundational YOCO architecture was noted as a key missing baseline. Concerns were also raised regarding the paper's clarity and the novelty of its contributions in the context of prior work. Overall though the consensus was that the strengths of the paper outweigh the limitations, and therefore I recommend accepting this paper.